# UNRAVELING INDIRECT IN-CONTEXT LEARNING USING INFLUENCE FUNCTIONS

## ABSTRACT

In this work, we introduce a novel paradigm for generalized In-Context Learning (ICL), termed *Indirect In-Context Learning*. In Indirect ICL, we explore demonstration selection strategies tailored for two distinct real-world scenarios: *Mixture of Tasks* and *Noisy ICL*. We systematically evaluate the effectiveness of Influence Functions (IFs) as a selection tool for these settings, highlighting the potential of IFs to better capture the informativeness of examples within the demonstration pool. For the Mixture of Tasks setting, demonstrations are drawn from 28 diverse tasks, including MMLU, BigBench, StrategyQA, and CommonsenseQA. We demonstrate that combining BertScore-Recall (BSR) with an IF surrogate model can further improve performance, leading to average absolute accuracy gains in 3-shot and 5-shot setups when compared to traditional ICL metrics. In the Noisy ICL setting, we examine scenarios where demonstrations might be mislabeled or have adversarial noise. Our experiments show that reweighting traditional ICL selectors (BSR and Cosine Similarity) with IF-based selectors boosts accuracy on noisy GLUE benchmarks. For the adversarial sub-setting, we show the utility of using IFs for task-agnostic demonstration selection for backdoor attack mitigation. Showing a reduction in Attack Success Rate compared to task-aware methods. In sum, we propose a robust framework for demonstration selection that generalizes beyond traditional ICL, offering valuable insights into the role of IFs for Indirect ICL.

## 1 INTRODUCTION

In-Context Learning (ICL) has emerged as a powerful method for utilizing large language models (LLMs) to handle novel tasks at inference (Mann et al., 2020; Min et al., 2022). Unlike traditional approaches that require task-specific fine-tuning, ICL allows a single model to adapt to different tasks without additional training, relying solely on the demonstrations provided in the context. This flexibility not only reduces the cost of task adaptation but also offers a transparent and easily customizable way of guiding the model's behavior (Liu et al., 2021a; Wei et al., 2022). By leveraging the context provided in prompts, ICL has been shown to improve both generalization across diverse tasks and reasoning abilities (Anil et al., 2022; Drozdov et al., 2022). Despite its advantages, the success of ICL is closely tied to the choice of demonstrations used in the prompt. Even slight variations in these demonstrations can significantly influence the model's performance, as shown in numerous studies (Zhao et al., 2021; Liu et al., 2021a; Lu et al., 2022).

Traditional ICL makes numerous assumptions that restrict its applicability to real-world problem domains. For instance, traditional ICL (Min et al., 2021; Conneau, 2019; Halder et al., 2020) assumes that demonstrations to be selected are *directly* and accurately annotated for the end-task. However, this is not always the case – for low-resource, sparse, or specialized domains, end-task information and labeled demonstrations might not be available. Similarly, when LLMs are deployed as services, the user query or the end task itself could be unknown beforehand, let alone providing direct demonstrations at inference.[1] Thus, in this paper, we explore a more generalized setting for ICL, which we refer to as ***Indirect ICL***.

---

[1]Our proposed method can improve performance by selecting relevant demonstrations from a task agnostic pool of labeled data at test time.

Indirect ICL appears in real-world applications in several ways. For instance, it can be applied to diagnosing rare medical conditions, where no precedents or labeled demonstrations exist. Similarly, it is relevant for niche programming languages or indigenous spoken languages, which often lack sufficient labeled data. In Indirect ICL, we aim to provide indirect (or incidental) supervision (Yin et al., 2023; Li et al., 2024) by selecting demonstrations from a pool of examples where the majority are not directly suited to the end task due to severe distribution/covariate shifts. This includes selecting demonstrations from a pool that predominantly consists of demonstrations belonging to other tasks, with few demonstrations from the end task possibly included. Additionally, the demonstration set may be mislabeled by humans (Yan et al., 2014; Zhu et al., 2022) or LLMs (Wu et al., 2023). Since the effectiveness of ICL heavily relies on the quality of demonstrations selected (Kossen et al., 2024; Wu et al., 2022; Wang et al., 2024), selecting the most helpful indirect demonstrations becomes imperative in these situations. We provide detailed examples of practical applications of Indirect ICL in Appendix A.

Despite these potential issues with the demonstration set, we wish to pave the way for extracting maximal benefit from any type of annotated dataset, irrespective of label purity or task-relatedness. Moreover, existing approaches designed for Direct ICL often fail to generalize effectively to the Indirect ICL setting. For example, in an ablation study reported in Appendix C.7, we observe that standard retrieval methods such as Cosine Similarity and BERTScore-Recall select demonstrations from related tasks only 33.90% and 36.03% of the time, respectively, when evaluated on a pool where the vast majority of tasks are unrelated.

Thus, in order to combat the aforementioned issues with sub-optimal datasets for ICL and sub-optimal demonstration selection strategies, we leverage *Influence Functions (IFs)* (Hampel, 1974; Cook & Weisberg, 1980). IFs offer a formal method for assessing how individual training data points affect model predictions. They have proven effective in a range of downstream data-centric learning tasks, including mislabeled data detection (Koh & Liang, 2017; Pruthi et al., 2020), optimal subset selection (Feldman & Zhang, 2020; Guo et al., 2020; Xia et al., 2024), model interpretation (Han et al., 2020; Grosse et al., 2023; Chhabra et al., 2024), data attribution (Bae et al., 2024), data valuation (Choe et al., 2024) and analyzing model biases (Wang et al., 2019; Kong et al., 2021).

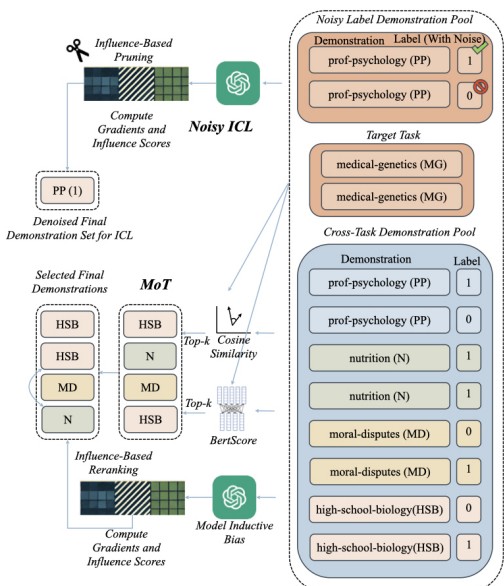

Figure 1: Example showcasing demonstration selection for Indirect ICL using Influence Functions (IFs). Consider web corpora with many tasks (different from the end-task) and noisy data– Indirect ICL can be formalized as: *Mixture of Tasks* (Section 3.1) and *Noisy* (Section 3.2) ICL, respectively. In MoT, for a given target task (e.g. *Medical Genetics*), we first filter from this (indirect) pool of candidate demonstrations using BertScore and Cosine Similarity, then re-rank with IFs to select suitable demonstrations (e.g. *High-School Biology*). For Noisy ICL, we leverage IFs to filter out the Noisy ICL before conducting ICL with the remaining clean demonstrations.

Traditional (direct) ICL methods that use metrics such as BertScore-Recall (BSR; Gupta et al. 2023a) and cosine similarity (Reimers, 2019) inherently rely on the semantic similarity between demonstrations and test samples. In this paper, we posit that IFs can be a reasonable measure of affinity between the end task and any (indirect) demonstrations. We show that it is practical to use IFs to identify candidate demonstrations that represent a close inductive bias with the end-task, and utilize this information for highly accurate demonstration selection in the challenging Indirect ICL setting. As our experiments and results will demonstrate, this is indeed the case, and we find that IFs can aid in improved performance when simple semantic similarity is insufficient for demonstration selection.

**Contributions.** In sum, our work advances ICL demonstration selection and makes the following key contributions and findings:

- We formalize a new and general paradigm for ICL, namely Indirect In-Context Learning, where we benchmark demonstration selection for two distinct and real-world settings: (a) **Mixture of Tasks** and (b) **Noisy ICL**. This novel paradigm with two settings is ubiquitous in the real world, and has yet been overlooked by existing research in ICL that assumes the availability of direct supervision.
- We propose utilizing Influence Functions (IFs) as an effective approach for demonstration selection in generalized ICL settings, leveraging their capacity to exploit the task inductive bias of models to enhance selection quality. We also examine multiple influence functions for Indirect ICL and conduct an extensive analysis on their benefits in this setting.
- For Mixture of Tasks, combining an IF Surrogate model with BertScore-Recall (BSR) can lead to an increase in performance for $k = 3$ and $k = 5$ shots compared to the best performing traditional ICL metric.
- For Noisy ICL, we observe that undertaking a weighted average selection using traditional ICL selectors (BSR and Cosine Similarity) and IF based selectors increases the absolute accuracy for mislabeled samples. IFs can also lead to a reduction in the Attack Success Rate for task-agnostic demonstration selection for backdoor attack mitigation.

## 2 PRELIMINARIES

We hereby introduce preliminaries of ICL and IF.

### 2.1 TRADITIONAL IN-CONTEXT LEARNING

Before we define the more generalized problem of Indirect ICL, we first define traditional ICL.

**In-Context Learning.** ICL allows LLMs to solve test inputs from novel tasks by presenting a few examples of the task in the prompt. Formally, given a set of input $x$ and output $y$ pairs $\{(x_i, y_i)\}_{i=1}^{k}$, prompt template $T$, and the test input $x_{\text{test}}$, ICL using an LLM involves prompting it to conditionally generate the test output $y_{\text{test}}$ according to the following distribution:

$$y_{\text{test}} \sim P_{\text{LM}}(\cdot \mid T(x_1, y_1, \ldots, x_k, y_k, x_{\text{test}}))$$

**Demonstration Selection.** In this work we study the problem of selecting $k$ in-context examples from a pool of $N \gg k$ labeled candidates. This is often necessary due to context length limits and cost considerations (Rubin et al., 2021; Gupta et al., 2023a). Formally, the goal is to select a subset $S \subset \{(x_i, y_i)\}_{i=1}^{N}$ of size $k$ that maximizes the probability of generating the desired $y_{\text{test}}$ when the LLM is conditioned on $x_{\text{test}}$ and $S$. It is noteworthy that prior studies mainly consider a task-dependent ICL scenario and assume that candidate demonstrations all directly match the end task (Min et al., 2021; Conneau, 2019; Halder et al., 2020).

### 2.2 INDIRECT IN-CONTEXT LEARNING

Now, we describe two scenarios of Indirect ICL, one where the candidate pool comprises of demonstrations from various tasks and the other where the demonstrations may have noisy labels.

**Mixture of Tasks.** Unlike traditional ICL, where candidate demonstrations match the end task at inference, we consider the more generalized Indirect ICL setting where the demonstration pool is task-agnostic. In practice, this setting would allow for pooling annotated demonstrations from various accessible tasks. Formally, given a set of input $x$ and output $y$ pairs $\{(x_i, y_i)\}_{i=1}^{k}$, where the pairs $(x_i, y_i)$ may originate from different tasks than the test input $x_{\text{test}}$, the model is prompted to maximize performance across test tasks.

**Noisy ICL.** To further generalize the problem of Indirect ICL, we also consider noisy supervision that is likely existing in the pool of demonstrations. Formally, let $D = \{(x_i, y_i)\}_{i=1}^{n}$ denote the training dataset, where $x_i \in X$ is the input and $y_i \in Y$ is the corresponding binary label. We randomly select a percentage of data points from $D$ and flip their labels. Once the noisy dataset is generated, we use it for ICL. Given the noisy set of input-output pairs $\{(x_i, y_i)\}_{i=1}^{k}$ and a test input $x_{\text{test}}$, the goal is to conditionally generate the test output $y_{\text{test}}$ based on the noisy training data.

## 2.3 INFLUENCE FUNCTIONS

Here we formally define how we will use IFs to perform Generalized Indirect ICL.

Let the input space be $X$ and the label space be $Y$. The training dataset is denoted as $D = \{(x_i, y_i)\}_{i=1}^n$, where $x_i \in X$ and $y_i \in Y$ are the input and label of the $i$-th data point. Given a loss function $\ell$ and a parameter space $\Theta$, the empirical risk minimization problem is defined as:

$$\theta^* = \arg\min_{\theta \in \Theta} \frac{1}{n} \sum_{i=1}^n \ell(y_i, f_\theta(x_i)),$$

where $f_\theta : X \to Y$ is the model parameterized by $\theta \in \Theta$. The gradient of the loss for the $i$-th data point with respect to a vector $\eta$ is denoted as:

$$\nabla_\eta \ell_i = \nabla_\eta \ell(y_i, f_\theta(x_i)).$$

The IF evaluates the effect of individual training data points on the estimation of model parameters (Hampel, 1974; Cook & Weisberg, 1980; Martin & Yohai, 1986). It measures the rate at which parameter estimates change when a specific data point is up-weighted.

Specifically, for $k \in [n]$ and $\epsilon \in \mathbb{R}$, we consider the following $\epsilon$-weighted empirical risk minimization problem:

$$\theta^{(k)}(\epsilon) = \arg\min_{\theta \in \Theta} \frac{1}{n} \sum_{i=1}^n \ell(y_i, f_\theta(x_i)) + \epsilon \ell(y_k, f_\theta(x_k)).$$

Here, the loss function $\ell(y, f_\theta(x))$ is assumed to be twice-differentiable and strongly convex in $\theta$ for all $(x, y) \in \mathcal{X} \times \mathcal{Y}$, the empirical risk minimizer (model weights) $\theta^*$ is well-defined, and the influence of the $k$-th data point $(x_k, y_k) \in D$ on the empirical risk minimizer (model weights) $\theta^*$ is defined as the derivative of $\theta^{(k)}(\epsilon)$ at $\epsilon = 0$:

$$I_{\theta^*}(x_k, y_k) := \left. \frac{d\theta^{(k)}}{d\epsilon} \right|_{\epsilon=0} = -H(\theta^*)^{-1} \nabla_\theta \ell(y_k, f_\theta(x_k)).$$

where $H(\theta) := \nabla_\theta^2 \left( \frac{1}{n} \sum_{i=1}^n \ell(y_i, f_\theta(x_i)) \right)$ is the Hessian of the empirical loss.

The IF $I_{\theta^*}(x_k, y_k)$ on the empirical risk minimizer $\theta^*$ is generalized to assess its effect on prediction loss (Koh & Liang, 2017). Given a validation dataset $D^\mathcal{V} := \{(x_i^\mathcal{V}, y_i^\mathcal{V})\}_{i=1}^m$, the influence of $(x_k, y_k)$ on the validation loss is defined as:

$$I(x_k, y_k) := \left( \frac{1}{m} \sum_{i=1}^m \nabla_\theta \ell(y_i^\mathcal{V}, f_\theta(x_i^\mathcal{V})) \Big|_{\theta=\theta^*} \right)^\top \times I_{\theta^*}(x_k, y_k).$$

This gives us

$$I(x_k, y_k) = -\sum_{i=1}^m \left( \nabla_\theta \ell(y_i^\mathcal{V}, f_\theta(x_i^\mathcal{V}))^\top H(\theta^*)^{-1} \nabla_\theta \ell(y_k, f_\theta(x_k)) \right).$$

The IF $I(x_k, y_k)$ provides insight into how a single data point impacts the validation loss. Essentially, it indicates whether $(x_k, y_k)$ contributes positively or negatively to the prediction loss. The more positive the influence value, the more it contributes to the loss decreasing, hence it is a beneficial data point to train the model.

**Remark.** As discussed above, IFs assume convexity of the loss function, which does not hold for LLMs and deep neural networks. Even though the IF formulations we employ in this paper (Kwon et al., 2023; Koh & Liang, 2017) make this underlying assumption, we find through empirical observations that for indirect ICL, they can work well. Circumventing the convexity assumption in IF is an ongoing area of research (Grosse et al., 2023; Chhabra et al., 2025) and our framework is flexible enough to accommodate any future IF variants.

## 3 PROPOSED APPROACH

In this section, we describe our approach to select demonstrations in both sub tasks.

### 3.1 SELECTING WITHIN MIXTURE OF TASKS

In this scenario, we develop influence-based methods for demonstration selection. Specifically, for each validation example, we compute influence values to identify the most impactful examples from a pool of training examples containing a mixture of tasks. Two approaches are employed to calculate these influence scores:

- A **surrogate-model** based method, where a lightweight surrogate model such as RoBERTa (Liu, 2019) is fine-tuned on the candidate demonstrations to compute influence.
- A **pretrained-gradient** based method where the samples are passed through the LLM itself. We then compute IFs using the extracted gradients.

Formally, for each validation example $(x_{\text{val}}, y_{\text{val}})$, we compute the influence of each training example $(x_i, y_i) \in D_{\text{train}}$, where $D_{\text{train}}$ is the set of the training examples. The influence score $I((x_i, y_i), (x_{\text{val}}, y_{\text{val}}))$ quantifies the effect of $(x_i, y_i)$ on the loss function evaluated at $(x_{\text{val}}, y_{\text{val}})^2$. Using these computed influence values, we select the top $k$ samples with the highest IF scores.

We compare two versions of computing the IF after extracting the gradient, DataInf (Kwon et al., 2023) and TracIn (Pruthi et al., 2020). DataInf uses an easy-to-compute closed-form expression, leading to better computational and memory complexities than other IF methods, more details in Appendix B. TracIn traces how the loss on the test point changes during the training process simply using an inner product of training and validation set gradients. Since it does not compute the Hessian matrix, it is faster than DataInf, but at the cost of lower estimation performance.

Additionally, we compare the influence-only methods described above with strong ICL baselines BertScore-Recall (BSR; Gupta et al. 2023a) and Cosine Similarity (Reimers, 2019). These methods excel at capturing semantic similarity between validation and training examples. We also compare with a performant sparse information retrieval baseline algorithm, BM25 (Jones et al., 2000).

Lastly, we combine the previously described approaches by implementing a two-stage selection process. First, we perform an initial pruning of the demonstration pool using either BSR or Cosine Similarity. Specifically, for a given number of desired demonstrations $k$, we prune the dataset to select $2k$ candidates from the original set of labeled examples $\{(x_i, y_i)\}_{i=1}^N$. We then apply the IF-based methods to re-rank these remaining examples based on their influence scores. The final selection of $k$ in-context demonstrations is performed by selecting the top $k$ examples from the re-ranked subset.

### 3.2 SELECTING NOISY ICL

In this setting, we utilize IFs to identify noisy mislabeled samples within the dataset. Formally, let $D = \{(x_i, y_i)\}_{i=1}^N$ represent the training dataset. First, we employ IFs to prune the dataset by detecting and removing noisy examples, following which the top $k$ in-context demonstrations are selected using either BSR or Cosine Similarity. We will refer to this approach as IF PRUNING.

Additionally, we construct approaches that combine the influence values with the BSR or Cosine Similarity scores. To do so, both the influence values and similarity scores are min-max normalized, resulting in scores scaled between 0 and 1. We then reweigh the scores using a linear combination of the normalized values. Let $\alpha$ and $\beta$ represent the weights assigned to the influence values and the similarity scores, respectively, where $\alpha + \beta = 1$ and $0 < \alpha, \beta < 1$, the final combined score for each training example is:

$$\text{Score}(x_i, y_i) = \alpha \cdot I((x_i, y_i), (x_{\text{val}}, y_{\text{val}})) + \beta \cdot S(x_i, y_i),$$

where $I((x_i, y_i), (x_{\text{val}}, y_{\text{val}}))$ is the influence value and $S(x_i, y_i)$ represents either the BertScore (Zhang et al., 2019) or Cosine Similarity for the training example $(x_i, y_i)$. The top $k$

---

$^2$We use the standard negative log-likelihood as the loss function.

examples with the highest combined scores are selected as demonstrations. We will refer to this approach as IF AVERAGING.

In this setting, we compute influence values using our surrogate model approach. In addition to using DataInf, we also conduct influence experiments using the LiSSA IF method which is a second-order method to compute the inverse Hessian vector product (Agarwal et al., 2017; Koh & Liang, 2017). Although LiSSA is generally computationally expensive (Kwon et al., 2023), we prioritize it over TracIn owing to its greater performance in detecting mislabeled samples, as the computational overhead is incurred only once in this setting[3].

## 4 EXPERIMENTS

Here, we expand upon our experimental setup to conduct the experiments and analyze the results. Overall, we find that combining DataInf with BSR for demonstration selection consistently improves accuracy across both the Mixture of Tasks and Noisy ICL settings within the Indirect ICL framework. Additionally, the surrogate model approach outperforms the pretrained gradient approach.

### 4.1 EXPERIMENTAL SETUP

We discuss our dataset details and model used to conduct the experiments.

**Evaluation Data.** For *Mixture of Tasks*, we collect a generalized pool of examples from different tasks such that the input $x$ and output $y$ pairs $\{(x_i, y_i)\}_{i=1}^{k}$ do not necessarily correspond to the same task as the test input $x_{\text{test}}$. The evaluation task pool contains three samples each from 28 different tasks from MMLU (Hendrycks et al., 2020), BigBench (Srivastava et al., 2022), StrategyQA (Geva et al., 2021) and CommonsenseQA (Talmor et al., 2018). We evaluate the ICL accuracy, using this train set, on 12 different tasks from MMLU and BigBench.

For *Noisy ICL*, we employ the noisy dataset framework from Kwon et al. (2023). In their work, the four binary classification GLUE datasets (Wang, 2018) MRPC, QQP, QNLI, and SST2 are utilized. To simulate a scenario where a portion of the data samples are noisy, 20% of the training data samples are randomly selected and their labels are flipped. We use these noisy datasets as the candidate pool in our experiments and evaluate the ICL accuracy.

**Base LLM.** In Mixture of Tasks, for $k = 3$ shots, we conduct ICL experiments on Llama-2-13b-chat (Touvron et al., 2023), Mistral-7b-v0.3 (Jiang et al., 2023), Zephyr-7b-beta (Tunstall et al., 2023), Qwen2.5-3b (Team, 2024) and Llama-3-70b (Grattafiori et al., 2024). For $k = 5$ shots we conduct experiments on Llama-2-13b-chat. We extend on the framework designed by Gupta et al. (2023a;b). The temperature is set to 0 for inference. For Noisy ICL, we conduct experiments on Llama-2-13b-chat for mislabeled data detection and Llama-3-8b for backdoor defense. All of our experiments run on $8\times$NVIDIA RTX 6000 Ada GPUs.

### 4.2 METHOD AND BASELINE CONFIGURATIONS

Here we expand on the methods and baselines we use for our experiments in both settings.

**Mixture of Tasks.** We construct 4 IF-only methods. 2 based on the Surrogate Model based approach, SUR and 2 based on the Pretrained LLM weights based approach, PRE. We test Data-Inf and TracIn based versions of these approaches, namely, Surrogate Model-DataInf SUR$_D$, Surrogate Model-TracIn SUR$_T$, Pretrained Model-DataInf PRE$_D$ and Pretrained Model-TracIn PRE$_T$. As mentioned before, SUR$_D$ and SUR$_T$ use RoBERTa as the surrogate model, whereas PRE$_D$ and PRE$_T$ use Llama2-13b-chat as the pretrained LLM. Additionally, we test traditional semantic approaches, such as BSR and Cosine Similarity (COS), as well as retrieval based approaches, such as BM25, as baselines. Finally, we test the combination of the aforementioned traditional and IF methods as well.Specifically, we denote these as:

$\text{MODEL}_{[\text{IF,SEL}]}$, where $\text{MODEL} \in \{\text{SUR}, \text{PRE}\}$, $\text{IF} \in \{\text{D}, \text{T}\}$, and $\text{SEL} \in \{\text{COS}, \text{BSR}\}$.

---

[3]We use different influence algorithms since Noisy ICL computes IF scores using all validation samples, enabling detection of noisy training samples. In contrast, MoT requires sample-specific IF scores, necessitating dynamic and per-instance computation.

**Noisy ICL.** As elaborated in Section 3.2, we explore two approaches, IF Pruning and IF Averaging, for the task of selecting the best demonstrations. We only use the surrogate model-based IF method in this setting, and we employ an additional method of computing IFs, LiSSA (Koh & Liang, 2017). We experiment with different levels of pruning and IF weights ($\alpha$) as hyperparameters, namely 10% pruning and $0.5\alpha$. Furthermore, we also create a random pruning variation for BSR and Cosine Similarity as well. Formally, we denote these as $\textbf{METHOD}[\textbf{MODEL}_{\textbf{IF,SEL}}]$, where $\textbf{METHOD} \in \{\textbf{PRU}, \textbf{AVG}\}$, $\textbf{MODEL} \in \{\textbf{SUR}, \textbf{RAND}\}$, $\textbf{IF} \in \{\textbf{D}, \textbf{L}\}$, and $\textbf{SEL} \in \{\textbf{COS}, \textbf{BSR}\}$.

### 4.3 RESULTS ON MIXTURE OF TASKS

We present the results on Mixture of Tasks in Figure 2. Additional results for varying the number of shots ($k$) and multiple LLMs are provided in Appendix C.1. Further, results for the alternative TracIn IF method are provided in Appendix C.2. Results for Pretrained Gradients combined with BertScore and Cosine Similarity are presented in Appendix C.3. We also present results on using DeBERTa (He et al., 2021a;b) as an alternative surrogate model in Appendix C.4. Finally, we present results on Qwen2.5-3b (Team, 2024) in Appendix C.5 and results on Llama-3-70b (Grattafiori et al., 2024) in Appendix C.6.

Figure 2: Average performance of different demonstration selection methods across Llama2-13b-chat, Zephyr-7b-beta and Mistral-7b-v0.3 for $k = 3$ shots.

**Combining Surrogate Model DataInf with BertScore results in the best performance.** As can be observed in Figure 2 and Table 6, the $\text{SUR}_{[D,BSR]}$ method has the highest average performance across the tasks, in both 3 and 5 shots. This shows the benefit of combining IF with BertScore as performance increased by 0.56 in $k = 3$ shots and by 1.52 in $k = 5$ shots. The results also show that the maximal benefit of IF methods is gained in combination with the semantic similarity methods. This is due to the fact that IF can leverage the model's inductive bias to re-rank the retrieved demonstrations effectively, but the initial $2k$ pruning via BSR is critical to shorten the candidate pool to demonstrations that are semantically relevant enough. However, it is important to note that, given the inclusion of only three shots in the prompt, where the overwhelming majority of demonstrations are unrelated to the test task, achieving significant improvements remains challenging. We provide an ablation diving deeper into why combining IF with semantic similarity methods works well in Appendix C.7.

**Surrogate Models outperform Pretrained Gradients.** We see that surrogate models outperformed Pretrained Gradients in demonstration selection for the Mixture of Tasks setting in both the $k = 3$ and $k = 5$ shots. The fine-tuning of the surrogate model leads it to better capture the test task affinity of the demonstration pool. However, it is interesting to note that on more recent LLMs Appendix C.5 and LLMs with larger number of parameters Appendix C.6 PreTrained Gradients outperformed the surrogate models.

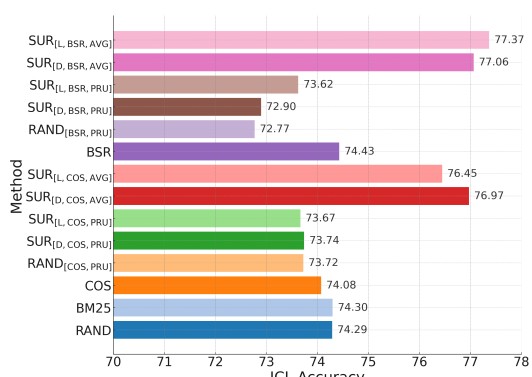

Figure 3: Average performance of the baselines across the 4 datasets.

**DataInf is better than TracIn as an IF method.** The speed gains of TracIn come at a cost of performance as the DataInf method of IF computation routinely outperformed TracIn. TracIn likely underperforms because it does not utilize critical second order gradient information

since the Hessian $H(\theta^*)$ is assumed to be the identity matrix. This trend has also been observed in past work on IF methods (Chhabra et al., 2025).

**Qualitative Analysis.** Finally, to understand the unique benefits provided by IFs, we present a qualitative analysis examining the types of shots selected by our method in Appendix C.8. We see that even though BSR selects more semantically relevant samples, $\text{SUR}_{[D,BSR]}$ shots assist in guiding the model toward the correct answer by providing examples that promote more structured reasoning.

## 4.4 RESULTS ON NOISY ICL

In this section, we present results for the mislabeled data setting under Noisy ICL. Additionally, we report results for adversarial noise, specifically a backdoor defense strategy, in Appendix H.

**IF Averaging works better than other baselines.** Table 1 and Figure 3 clearly show that doing a weighted average between the surrogate model IF and both Cosine and BertScore leads to performance boosts. Atleast one and if not both of the highest performing methods in each of the datasets we tested were from the averaging method. We see that LiSSA and DataInf are similarly effective, with DataInf being more computationally efficient.

**Pruning hurts not helps performance.** We can see that pruning actively hurts performance as Figure 3 shows that all 3 types of BertScore pruning and all 3 types of Cosine pruning had lower average scores than BertScore and Cosine Similarity. This might be due to the fact that we are removing potentially helpful samples from the demonstration pool, even if they might have noisy labels. We further provide results on varying the noise levels in the datasets in D.1, varying the hyperparameters we tested in D.2 and an experiment analyzing the effectiveness of IF's in detecting Noisy ICL in D.3.

Table 1: ICL Accuracy across MRPC, QNLI, SST2, and QQP datasets using different methods for Noisy ICL, with 20% noise added to the datasets. The top 2 performers for each dataset are in bold.

| | Method | MRPC | QNLI | SST2 | QQP |
|---|---|---|---|---|---|
| | **RAND** | 70.4 | 69.6 | 86.2 | 70.9 |
| | **BSR** | 71.3 | 74.6 | 80.4 | 71.4 |
| | **COS** | 72.3 | 68.2 | 82.6 | **73.2** |
| | **BM25** | 70.6 | 67.6 | 88.0 | 71.0 |
| PRU-0.1 | **RAND**[Cos] | 70.1 | 68.4 | 87.0 | 69.4 |
| | **SUR**[D,Cos] | 69.4 | 69.8 | 84.0 | 71.8 |
| | **SUR**[L,Cos] | 68.9 | 68.2 | 86.8 | 70.8 |
| | **RAND**[BSR] | 71.1 | 65.2 | 86.4 | 68.4 |
| | **SUR**[D,BSR] | 70.6 | 68.0 | 82.0 | 71.0 |
| | **SUR**[L,Cos] | 70.1 | 67.4 | 88.8 | 68.2 |
| AVG-0.5 | **SUR**[D,Cos] | **75.5** | **74.8** | 89.8 | 67.8 |
| | **SUR**[L,Cos] | 70.6 | **75.8** | 86.4 | 73.0 |
| | **SUR**[D,BSR] | **74.3** | 69.6 | **90.6** | **73.8** |
| | **SUR**[L,BSR] | 73.3 | 73.4 | **93.6** | 69.2 |

**IF protect against adversarial noise.** The results presented in Appendix H show that the IF-based indirect-ICL paradigm mitigates backdoor attacks, reducing ASR by 32.89% on average. Even without task-specific data, demonstrations guided by a model's inductive bias offer a strong task-agnostic backdoor defense.

## 4.5 COMPUTATIONAL COMPLEXITY

We present the worst case time complexity (for inference) for our methods and related baselines in Table 2. As can be observed, our methods are comparable, if not more efficient than the other baselines. Note that the SUR methods require an additional fine-tuning step on a smaller surrogate model before the gradients are extracted, which

Table 2: Computational complexity for each test sample at inference. $N$ is #demonstration samples, $p$ is #model parameters, $d$ is embedding size, $K$ is the max candidate length, $L$ is the length (in tokens) of the test input, $Z$ is #ngrams

| Method | Time Complexity |
|---|---|
| **BSR** | $O(NLKd)$ |
| **COS** | $O(Nd)$ |
| **BM25** | $O(NZ)$ |
| **PRE**$_D$ | $O(Np)$ |
| **SUR**$_D$ | $O(Np)$ |
| **PRE**$_T$ | $O(Np)$ |
| **SUR**$_T$ | $O(Np)$ |
| **BSR COMBINED METHODS** | $O(NLKd) + O(Np)$ |
| **COS COMBINED METHODS** | $O(Nd) + O(Np)$ |

the PRE methods do not. Furthermore, note that TracIn as an influence method is much faster than Hessian-based approaches (e.g. DataInf) as it assumes that the Hessian is the identity matrix. While this leads to more efficient influence computation, it comes at the cost of lower estimation performance, as our results with TracIn also show. Additionally, we present the maximum GPU memory

consumption while performing demonstration selection in Appendix E, scalability to large models in Appendix F and scalability to large datasets in Appendix G. The experiments demonstrate the feasibility of applying our methods at increasing scales. Nevertheless, it is important to recognize that improving the efficiency of IFs remains an active research area, and such advancements can be readily integrated into our approach to further enhance the efficiency and accuracy of Indirect-ICL performance.

## 5 RELATED WORK

**In-Context Learning (ICL).** Following the scaling of model sizes and learning resources (Mann et al., 2020; Chowdhery et al., 2023; Touvron et al., 2023), LLMs have gained emergent abilities for efficient inference-time adaptation via ICL (Mann et al., 2020). However, ICL is critically sensitive to demonstration pool examples (An et al., 2023; Liu et al., 2021a; Zhang et al., 2022) and selection strategies (Rubin et al., 2021; Mavromatis et al., 2023). One line of work studies example scoring and retrieval, utilizing model-agnostic heuristic metrics like perplexity (Gonen et al., 2022), mutual information (Sorensen et al., 2022), semantic similarity (Liu et al., 2021a; Gupta et al., 2023b), etc. to select demonstrations. Another line of work optimizes selection based on empirically verified desirable features a priori, e.g. diversity (Su et al., 2022; Ye et al., 2023), coverage (Gupta et al., 2023a), etc. However, prior work assumes that the demonstration distribution is aligned with task distribution, which is not always the case (Chatterjee et al., 2024). Our work serves as a first to investigate ICL demonstration selection in the task and dataset quality shifts in the ICL settings.

**Influence Functions.** *Influence functions* (IFs) comprise a set of methods from robust statistics (Hampel, 1974; Cook & Weisberg, 1982) that have been recently proposed for deep learning data valuation and can provide a conceptual link that traces model performance to samples in the training set. For gradient-based models trained using empirical risk minimization, IFs can be used to approximate sample influence without requiring actual leave-one-out retraining. For deep learning models, the seminal work by Koh & Liang (2017) utilized a Taylor-series approximation and LiSSA optimization (Agarwal et al., 2017) to compute sample influences. Follow-up works such as Representer Point (Yeh et al., 2018) and Hydra (Chen et al., 2021) sought to improve IF performance for deep learning models, constrained to vision applications. More recently, efficient influence estimation methods such as DataInf (Kwon et al., 2023), Arnoldi iteration (Schioppa et al., 2022), and Kronecker-factored approximation curvature (Grosse et al., 2023) have been proposed which can be employed for larger generative language models, such as LLMs. Some other simpler IF approaches simply consider the gradients directly as a measure of influence (Pruthi et al., 2020; Charpiat et al., 2019), followed by some ensemble strategies (Bae et al., 2024; Kim et al., 2024). Recent work has also found that *self-influence* only on the training set can be a useful measure for detecting sample influence (Bejan et al., 2023; Thakkar et al., 2023).

IFs have been utilized with great success in a number of application scenarios (e.g. classification (Chhabra et al., 2025; Koh & Liang, 2017), generative models (Kwon et al., 2023; Schioppa et al., 2022; Grosse et al., 2023), active learning (Chhabra et al., 2024; Liu et al., 2021b), layer-quality estimation (Askari et al., 2025), etc.). Moreover, while some recent works have considered using influence for selecting direct demonstrations (Nguyen & Wong, 2023; Van et al., 2024), neither of them has considered their effect on inductive bias selection in the indirect ICL setting, which is the focus of our work.

## 6 CONCLUSION

We formalize a new paradigm for generalized In-Context Learning, which we term *Indirect In-Context Learning*. We analyze two different real-world Indirect ICL settings and propose effective demonstration selection strategies for these scenarios. We explore using Influence Functions (IFs) to leverage the informativeness of the samples in the demonstration pool and the models' task inductive bias. We find that combining a surrogate model-based IF approach with BertScore performs better when there are an overwhelming majority of irrelevant tasks in the candidate pool. We also find that reweighting the surrogate model-based IF scores with traditional metric scores can be helpful in the case of Noisy ICL. Future work will aim to augment the Pretrained Gradient approach by finetuning the LLMs.

## REPRODUCIBILITY STATEMENT

We use the implementation of Gupta et al. (2023a) for BERTScore-Recall, cosine similarity, and BM25 retrieval. Cosine similarity is computed via dense retrieval using *all-mpnet-base-v2* from SentenceBERT, while BM25 uses the Okapi variant from the `rank_bm25` library (Brown, 2018). BERTScore-Recall employs *deberta-large-mnli* as the encoder. For the Random baseline, we report the average over 5 runs with different seeds. The temperature for all LLMs was set to 0 for inference. All of our experiments run on $8\times$NVIDIA RTX 6000 Ada GPUs. The surrogate model is augmented with rank-2 LoRA adapters on the value projections and fine-tuned for 10 epochs. Training uses cross-entropy loss, AdamW optimizer with a learning rate of 3*e-4 and a 6% warm-up, linear-decay schedule. The dataset used for finetuning is the training set demonstrations we want to compute the Influence scores for. An anonymized version of the source code and datasets are available at `https://anonymous.4open.science/r/IncontextInfluence-ICLR-40B7`

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

APPENDIX

## A  INTEGRATION TO PRACTICAL WORKFLOWS

We provide several use cases of Indirect ICL in real-world scenarios. Here are a few examples:

## A.1 PRACTICAL APPLICATIONS OF INDIRECT ICL

- **Enhancing prompt performance at test time:** If an LLM service provider needs to use in-context learning (ICL) to improve prompt performance during test time, they may not know the precise task beforehand or during inference (e.g., a novel task requested by a user in real-time). Indirect ICL and our proposed methods can improve performance by selecting relevant demonstrations from a task-agnostic pool of labeled data (i.e., the MoT setting), ensuring the model can adapt to various scenarios even when task-specific labeled samples (direct supervision) are not available.

- **Medical diagnosis:** Indirect ICL can be used to diagnose rare medical conditions based on symptoms. Since such conditions are rare, demonstrations for these specific cases are often unavailable. However, the model can learn diagnostic reasoning patterns from more common conditions with overlapping symptoms, improving accuracy for the rare cases.

- **Code generation for obscure programming languages:** Indirect ICL can aid in generating code for rarely-used or proprietary programming languages. Demonstrations from code generation tasks in related languages with similar structures can be leveraged, enabling the model to generalize and perform well in these low-resource scenarios.

- **Ideology Estimation from Underrepresented Contexts:** We can use our paradigm to estimate political ideology, or any other sort of text classification, from text in an underrepresented cultural or linguistic context. We can use demonstrations from ideology estimation in well-represented contexts such as Western political texts. The can transfer learned associations between linguistic cues and ideological stances, adapting them to the new context.

These examples highlight just a few of the practical applications of indirect ICL, particularly in low-resource settings.

## B DATAINF METHOD OF COMPUTING INFLUENCE FUNCTIONS

DataInf is an efficient method for estimating influence functions in deep neural networks, especially in parameter-efficient fine-tuning settings such as LoRA. Traditional influence function computation requires inverting large Hessian matrices, which is impractical for large models. DataInf addresses this by introducing a closed-form approximation that significantly reduces both computational and memory costs.

The key approximation is to swap the order of the matrix inversion and the average calculations as below:

$$\left( \frac{1}{n} \sum_{i=1}^{n} \nabla_{\theta_\ell} \ell_i \nabla_{\theta_\ell} \ell_i^\top + \lambda_\ell I_{d_\ell} \right)^{-1} \approx$$

$$\frac{1}{n} \sum_{i=1}^{n} \left( \nabla_{\theta_\ell} \ell_i \nabla_{\theta_\ell} \ell_i^\top + \lambda_\ell I_{d_\ell} \right)^{-1}$$

Using the Sherman-Morrison formula, the inverse in each term can be computed analytically, allowing a closed-form estimate of the influence:

$$\mathcal{I}_{\mathrm{D}}(x_k, y_k) = \sum_{\ell=1}^{L} \frac{1}{\lambda_\ell} \left( \frac{1}{n} \sum_{i=1}^{n} \frac{L_{\ell,i}}{\lambda_\ell + L_{\ell,ii}} L_{\ell,ik} - L_{\ell,k} \right)$$

This formulation enables DataInf to scale to large models by avoiding iterative solvers and full Hessian storage, making it practical for real-world LLM applications.

## C INDIRECT ICL RESULTS FOR LLM BASED INFLUENCE

### C.1 FULL RESULTS FOR MIXTURE OF TASKS

Following are the full results for the Mixture of Tasks setting. For $k = 3$ shots in Tables 3, 4, and 5. For $k = 5$ shots in Table 6.

Table 3: Performance across different datasets and demonstration selection methods with $k = 3$ shots. The datasets are sampled from sub-tasks of the MMLU and BigBench datasets for Llama-2-13b-chat.

| Dataset | RAND | BSR | COS | BM25 | PRE$_D$ | SUR$_D$ | SUR$_{[D,BSR]}$ | SUR$_{[D,Cos]}$ |
|---|---|---|---|---|---|---|---|---|
| medical-genetics | 80.60 | 87.00 | 84.00 | 79.00 | 76.00 | 81.00 | 86.00 | 82.00 |
| prof-psychology | 68.30 | 70.00 | 73.00 | 65.50 | 65.00 | 68.50 | 72.00 | 68.50 |
| formal-logic | 60.16 | 59.52 | 60.32 | 58.73 | 61.11 | 59.52 | 57.14 | 55.56 |
| moral-disputes | 81.00 | 78.00 | 79.50 | 80.50 | 78.50 | 76.00 | 80.50 | 76.50 |
| public-relations | 72.18 | 79.09 | 80.91 | 73.64 | 71.82 | 75.45 | 79.09 | 79.09 |
| comp-security | 76.80 | 76.00 | 80.00 | 76.00 | 76.00 | 80.00 | 76.00 | 76.00 |
| astronomy | 80.26 | 80.26 | 78.95 | 80.92 | 74.34 | 79.61 | 80.26 | 78.95 |
| abstract-algebra | 57.00 | 58.00 | 62.00 | 55.00 | 57.00 | 47.00 | 72.00 | 72.00 |
| nutrition | 75.50 | 77.50 | 79.00 | 78.00 | 77.50 | 77.00 | 79.50 | 81.50 |
| high-school-biology | 76.70 | 76.50 | 78.00 | 76.50 | 73.50 | 79.00 | 80.50 | 76.50 |
| formal-fallacies | 47.25 | 52.00 | 50.00 | 49.50 | 47.00 | 56.50 | 47.50 | 50.00 |
| tracking-3 | 40.20 | 44.00 | 39.00 | 45.00 | 40.00 | 37.00 | 43.50 | 39.00 |
| Average | 68.00 | 69.82 | **70.39** | 68.19 | 66.48 | 68.04 | **71.16** | 69.63 |

Table 4: Performance across different datasets and demonstration selection methods with $k = 3$ shots. The datasets are sampled from sub-tasks of the MMLU and BigBench datasets for Mistral-7b-v3

| Dataset | RAND | BSR | COS | BM25 | PRE$_D$ | SUR$_D$ | SUR$_{[D,BSR]}$ | SUR$_{[D,Cos]}$ |
|---|---|---|---|---|---|---|---|---|
| medical-genetics | 88.25 | 88.00 | 91.00 | 89.00 | 88.00 | 86.00 | 87.00 | 89.00 |
| prof-psychology | 84.50 | 84.00 | 84.50 | 82.00 | 85.00 | 83.00 | 85.50 | 84.00 |
| formal-logic | 66.47 | 66.67 | 69.05 | 65.08 | 63.49 | 66.67 | 68.25 | 69.05 |
| moral-disputes | 87.00 | 85.00 | 85.50 | 87.00 | 88.5 | 86.00 | 87.00 | 87.50 |
| public-relations | 83.41 | 82.73 | 81.82 | 84.55 | 84.55 | 83.64 | 84.55 | 81.82 |
| comp-security | 87.25 | 83.00 | 89.00 | 88.00 | 90.00 | 88.00 | 86.00 | 90.00 |
| astronomy | 87.34 | 88.82 | 89.47 | 88.16 | 84.87 | 86.18 | 86.18 | 86.18 |
| abstract-algebra | 57.75 | 63.00 | 60.00 | 60.00 | 59.00 | 50.00 | 64.00 | 64.00 |
| nutrition | 84.75 | 86.50 | 87.50 | 83.00 | 83.00 | 83.50 | 88.50 | 87.00 |
| high-school-biology | 85.63 | 84.00 | 86.50 | 85.00 | 87.00 | 84.00 | 87.50 | 87.00 |
| formal-fallacies | 49.63 | 53.50 | 50.00 | 53.50 | 48.00 | 46.50 | 52.50 | 53.50 |
| tracking-3 | 46.38 | 49.00 | 49.00 | 49.50 | 46.50 | 39.00 | 48.50 | 45.50 |
| Average | 75.70 | 76.19 | **76.95** | 76.23 | 75.66 | 73.54 | **77.13** | 77.05 |

## C.2 TRACIN RESULTS

Here we provide results for the TracIn method of Influence Computation for $k = 3$ shots in Tables 7, 8, and 9. We also provide results for $k = 5$ shots in Table 10.

## C.3 PRETRAINED GRADIENT RESULTS

Here we provide results for pretrained gradients method of computing IF. These can be found in for $k = 3$ shots in Tables 11, 12 and 13 and for $k = 5$ shots in Table 14.

## C.4 RESULTS USING A DIFFERENT SURROGATE MODEL

We present results where DeBERTa-v3-Large replaces RoBERTa-Large as the surrogate model. Evaluated on Llama2-13B-Chat with $k = 3$ shots. We compare the best-performing baseline SUR$_{[D,BSR]}$ with BSR in Table 15.

The results indicate that the DeBERTa surrogate model outperforms BSR. However, it is important to note that, given the inclusion of only three shots in the prompt—where the overwhelming ma-

Table 5: Performance across different datasets and demonstration selection methods with $k = 3$ shots. The datasets are sampled from sub-tasks of the MMLU and BigBench datasets for Zephyr-7b-beta

| Dataset | RAND | BSR | COS | BM25 | PRE$_D$ | SUR$_D$ | SUR$_{[D,BSR]}$ | SUR$_{[D,Cos]}$ |
|---|---|---|---|---|---|---|---|---|
| medical-genetics | 79.50 | 80.00 | 77.00 | 76.00 | 78.00 | 82.00 | 76.00 | 78.00 |
| prof-psychology | 74.50 | 74.00 | 74.50 | 74.50 | 72.00 | 74.50 | 73.00 | 74.00 |
| formal-logic | 69.44 | 65.87 | 65.87 | 65.08 | 66.67 | 71.43 | 65.87 | 61.11 |
| moral-disputes | 77.63 | 78.50 | 78.00 | 76.50 | 75.00 | 78.00 | 78.50 | 77.00 |
| public-relations | 75.00 | 80.91 | 72.73 | 73.64 | 76.36 | 75.45 | 76.36 | 76.36 |
| comp-security | 76.00 | 73.00 | 77.00 | 78.00 | 75.00 | 77.00 | 79.00 | 79.00 |
| astronomy | 79.77 | 81.58 | 80.26 | 80.26 | 74.34 | 80.92 | 79.61 | 82.24 |
| abstract-algebra | 52.00 | 53.00 | 53.00 | 51.00 | 51.00 | 50.00 | 62.00 | 55.00 |
| nutrition | 75.63 | 77.00 | 79.50 | 75.50 | 74.00 | 74.50 | 75.50 | 77.50 |
| high-school-biology | 77.63 | 81.00 | 80.50 | 79.50 | 78.50 | 80.00 | 78.50 | 78.00 |
| formal-fallacies | 49.75 | 57.50 | 55.50 | 56.50 | 52.50 | 55.00 | 51.50 | 46.50 |
| tracking-3 | 49.25 | 49.50 | 49.50 | 51.50 | 52.50 | 45.00 | 49.50 | 49.50 |
| **Average** | 69.68 | **70.99** | 70.28 | 69.83 | 68.83 | 70.31 | **70.45** | 69.51 |

Table 6: Performance across different datasets and demonstration selection methods with $k = 5$ shots for Llama-2-13b-chat

| Dataset | RAND | BSR | COS | BM25 | PRE$_D$ | SUR$_D$ | SUR$_{[D,BSR]}$ | SUR$_{[D,Cos]}$ |
|---|---|---|---|---|---|---|---|---|
| medical-genetics | 80.00 | 86.00 | 81.00 | 81.00 | 84.00 | 80.00 | 84.00 | 83.00 |
| prof-psychology | 71.00 | 71.00 | 73.50 | 66.00 | 70.00 | 68.50 | 77.00 | 71.00 |
| formal-logic | 62.70 | 59.52 | 57.94 | 56.35 | 58.73 | 68.25 | 62.70 | 61.90 |
| moral-disputes | 79.50 | 77.50 | 81.00 | 81.00 | 82.00 | 81.00 | 81.50 | 81.50 |
| public-relations | 70.00 | 78.18 | 81.82 | 76.36 | 70.91 | 77.82 | 80.91 | 78.18 |
| comp-security | 75.00 | 78.00 | 82.00 | 77.00 | 76.00 | 77.00 | 81.00 | 77.00 |
| astronomy | 78.95 | 85.53 | 82.89 | 80.92 | 80.92 | 82.89 | 84.87 | 81.58 |
| abstract-algebra | 52.00 | 63.00 | 62.00 | 58.00 | 63.00 | 55.00 | 67.00 | 65.00 |
| nutrition | 71.50 | 81.00 | 80.00 | 78.00 | 76.00 | 78.00 | 79.00 | 81.00 |
| high-school-biology | 73.00 | 79.00 | 80.00 | 76.50 | 77.00 | 79.00 | 81.50 | 75.50 |
| formal-fallacies | 46.50 | 48.50 | 48.00 | 50.00 | 47.50 | 43.50 | 53.00 | 43.50 |
| tracking-3 | 36.50 | 49.00 | 47.00 | 46 | 42.50 | 52.00 | 42.00 | 39.00 |
| **Average** | 66.38 | 71.35 | **71.42** | 68.93 | 69.04 | 70.24 | **72.87** | 69.84 |

Table 7: Performance across different datasets and different TracIn Influence methods with $k = 3$ shots for Llama2-13b-chat

| Dataset | PRE$_T$ | SUR$_T$ | SUR$_{[T,BSR]}$ | SUR$_{[T,Cos]}$ |
|---|---|---|---|---|
| medical-genetics | 77.00 | 79.00 | 81.00 | 85.00 |
| prof-psychology | 67.00 | 66.00 | 69.50 | 70.00 |
| formal-logic | 56.35 | 61.11 | 59.52 | 59.52 |
| moral-disputes | 79.50 | 76.00 | 82.00 | 79.50 |
| public-relations | 73.64 | 72.73 | 76.36 | 77.27 |
| comp-security | 78.00 | 74.00 | 76.00 | 79.00 |
| astronomy | 80.26 | 75.66 | 82.24 | 79.61 |
| abstract-algebra | 57.00 | 61.00 | 67.00 | 63.00 |
| nutrition | 79.50 | 79.50 | 78.50 | 81.00 |
| high-school-biology | 76.50 | 76.00 | 79.00 | 75.00 |
| formal-fallacies | 46.50 | 42.50 | 47.50 | 43.00 |
| tracking-3 | 41.00 | 35.50 | 40.00 | 39.50 |
| **Average** | 67.69 | 66.58 | 69.89 | 69.28 |

Table 8: Performance across different datasets and different TracIn Influence methods with $k = 3$ shots for Mistral-7b-v0.3

| Dataset | $\text{PRE}_\text{T}$ | $\text{SUR}_\text{T}$ | $\text{SUR}_\text{[T,BSR]}$ | $\text{SUR}_\text{[T,Cos]}$ |
|---|---|---|---|---|
| medical-genetics | 88.00 | 85.00 | 87.00 | 88.00 |
| prof-psychology | 83.00 | 80.00 | 84.00 | 86.50 |
| formal-logic | 65.08 | 69.05 | 65.87 | 68.25 |
| moral-disputes | 87.50 | 84.50 | 84.50 | 89.00 |
| public-relations | 80.91 | 82.73 | 85.45 | 81.82 |
| comp-security | 87.00 | 83.00 | 86.00 | 87.00 |
| astronomy | 86.18 | 86.18 | 88.16 | 90.13 |
| abstract-algebra | 61.00 | 51.00 | 62.00 | 57.00 |
| nutrition | 85.00 | 83.00 | 88.50 | 86.00 |
| high-school-biology | 86.50 | 84.50 | 88.00 | 85.00 |
| formal-fallacies | 47.00 | 52.00 | 54.50 | 62.50 |
| tracking-3 | 44.00 | 42.00 | 35.00 | 38.00 |
| Average | 75.10 | 73.58 | 75.75 | 76.60 |

Table 9: Performance across different datasets and different TracIn Influence methods with $k = 3$ shots for Zephyr-7b-beta

| Dataset | $\text{PRE}_\text{T}$ | $\text{SUR}_\text{T}$ | $\text{SUR}_\text{[T,BSR]}$ | $\text{SUR}_\text{[T,Cos]}$ |
|---|---|---|---|---|
| medical-genetics | 76.00 | 77.00 | 74.00 | 80.00 |
| prof-psychology | 72.50 | 71.50 | 72.50 | 72.50 |
| formal-logic | 67.46 | 69.84 | 67.46 | 64.29 |
| moral-disputes | 77.50 | 75.50 | 76.00 | 77.50 |
| public-relations | 76.36 | 74.55 | 79.09 | 72.73 |
| comp-security | 77.00 | 78.00 | 76.00 | 75.00 |
| astronomy | 76.32 | 78.95 | 81.58 | 80.92 |
| abstract-algebra | 50.00 | 48.00 | 53.00 | 52.00 |
| nutrition | 75.50 | 77.00 | 75.50 | 78.00 |
| high-school-biology | 77.50 | 75.50 | 80.50 | 78.00 |
| formal-fallacies | 50.00 | 41.00 | 46.00 | 50.00 |
| tracking-3 | 50.50 | 48.00 | 49.50 | 42.50 |
| Average | 68.89 | 67.90 | 69.26 | 68.62 |

Table 10: Performance across different datasets and different TracIn Influence methods with $k = 5$ shots for Llama2-13b-chat.

| Dataset | $\text{PRE}_\text{T}$ | $\text{SUR}_\text{T}$ | $\text{SUR}_\text{[T,BSR]}$ | $\text{SUR}_\text{[T,Cos]}$ |
|---|---|---|---|---|
| medical-genetics | 82.00 | 78.00 | 83.00 | 81.00 |
| prof-psychology | 69.00 | 67.50 | 73.50 | 73.50 |
| formal-logic | 61.90 | 61.11 | 57.14 | 57.14 |
| moral-disputes | 81.00 | 81.00 | 82.50 | 81.50 |
| public-relations | 70.00 | 70.00 | 73.64 | 78.18 |
| comp-security | 75.00 | 76.00 | 77.00 | 76.00 |
| astronomy | 82.24 | 76.32 | 83.55 | 78.95 |
| abstract-algebra | 53.00 | 56.00 | 65.00 | 64.00 |
| nutrition | 77.50 | 75.50 | 79.00 | 79.50 |
| high-school-biology | 77.00 | 74.50 | 82.50 | 77.00 |
| formal-fallacies | 45.50 | 47.00 | 39.50 | 48.00 |
| tracking-3 | 40.50 | 41.50 | 46.50 | 36.50 |
| Average | 67.87 | 67.03 | 70.23 | 69.27 |

jority of demonstrations are unrelated to the test task—achieving significant improvements remains challenging.

Table 11: Performance across different datasets and Pre-training based demonstration selection methods ($k = 3$ shots) for Llama2-13b-chat.

| Dataset | PRE[D,BSR] | PRE[D,Cos] | PRE[T,BSR] | PRE[T,Cos] |
|---|---|---|---|---|
| medical-genetics | 85.00 | 80.00 | 86.00 | 84.00 |
| prof-psychology | 68.50 | 73.50 | 70.50 | 69.50 |
| formal-logic | 55.56 | 57.14 | 57.94 | 54.76 |
| moral-disputes | 80.50 | 80.00 | 78.50 | 82.00 |
| public-relations | 80.91 | 77.27 | 74.55 | 78.18 |
| comp-security | 75.00 | 79.00 | 80.00 | 76.00 |
| astronomy | 80.26 | 76.32 | 78.95 | 77.63 |
| abstract-algebra | 57.00 | 56.00 | 58.00 | 61.00 |
| nutrition | 81.00 | 80.00 | 78.00 | 80.50 |
| high-school-biology | 75.50 | 73.50 | 80.00 | 76.00 |
| formal-fallacies | 52.50 | 48.00 | 46.50 | 45.50 |
| tracking-3 | 48.50 | 47.50 | 37.00 | 38.00 |
| Average | 70.02 | 69.02 | 68.83 | 68.59 |

Table 12: Performance across different datasets and Pre-training based demonstration selection methods with $k = 3$ shots for Mistral-7b-v0.3.

| Dataset | PRE[D,BSR] | PRE[D,Cos] | PRE[T,BSR] | PRE[T,Cos] |
|---|---|---|---|---|
| medical-genetics | 86.00 | 88.00 | 86.00 | 88.00 |
| prof-psychology | 87.50 | 83.50 | 85.00 | 85.50 |
| formal-logic | 64.29 | 65.87 | 65.08 | 65.87 |
| moral-disputes | 86.00 | 85.00 | 85.00 | 85.50 |
| public-relations | 85.45 | 80.00 | 86.36 | 84.55 |
| comp-security | 82.00 | 88.00 | 85.00 | 89.00 |
| astronomy | 88.16 | 90.13 | 87.50 | 89.47 |
| abstract-algebra | 62.00 | 59.00 | 62.00 | 60.00 |
| nutrition | 86.50 | 84.00 | 86.50 | 84.50 |
| high-school-biology | 86.00 | 85.50 | 86.50 | 87.00 |
| formal-fallacies | 54.50 | 46.00 | 51.00 | 50.00 |
| tracking-3 | 49.50 | 48.50 | 50.00 | 53.00 |
| Average | 76.49 | 75.29 | 76.32 | 76.87 |

Table 13: Performance across different datasets and Pre-training based demonstration selection methods with $k = 3$ shots for Zephyr-7b-beta.

| Dataset | PRE[D,BSR] | PRE[D,Cos] | PRE[T,BSR] | PRE[T,Cos] |
|---|---|---|---|---|
| medical-genetics | 82.00 | 77.00 | 81.00 | 82.00 |
| prof-psychology | 73.00 | 70.50 | 74.50 | 73.00 |
| formal-logic | 69.84 | 67.48 | 65.08 | 66.67 |
| moral-disputes | 73.00 | 76.00 | 75.50 | 76.50 |
| public-relations | 78.18 | 77.27 | 76.36 | 70.00 |
| comp-security | 78.00 | 73.00 | 75.00 | 76.00 |
| astronomy | 76.97 | 78.29 | 77.63 | 79.61 |
| abstract-algebra | 51.00 | 55.00 | 58.00 | 58.00 |
| nutrition | 79.00 | 74.50 | 76.50 | 78.50 |
| high-school-biology | 78.00 | 77.00 | 80.00 | 78.50 |
| formal-fallacies | 54.50 | 55.50 | 46.50 | 54.50 |
| tracking-3 | 47.00 | 47.50 | 49.00 | 48.50 |
| Average | 70.04 | 69.08 | 69.59 | 70.15 |

## C.5 RESULTS ON QWEN2.5-3B

Here we present results on Qwen2.5-3b (Team, 2024) for $k = 3$ shots on the Mixture of Tasks setting. We compare BSR with our two IF based BSR methods SUR[D,BSR] and PRE[D,BSR]. We see in Table 16 that for an advanced model like Qwen2.5-3b, the unfinetuned pretrained gradients slightly outperform the finetuned gradients from the RoBERTa model.

Table 14: Performance across different datasets and Pre-training based demonstration selection methods with $k = 5$ shots for Llama2-13b-chat.

| Dataset | PRE[D,BSR] | PRE[D,Cos] | PRE[T,BSR] | PRE[T,Cos] |
|---|---|---|---|---|
| medical-genetics | **83.00** | **83.00** | 82.00 | **83.00** |
| prof-psychology | 73.50 | **74.00** | **74.00** | 71.50 |
| formal-logic | 60.32 | 56.35 | **62.70** | **62.70** |
| moral-disputes | 82.00 | **82.50** | 81.00 | 79.50 |
| public-relations | 75.45 | 75.45 | **78.18** | 77.27 |
| comp-security | 77.00 | 77.00 | 76.00 | **80.00** |
| astronomy | **82.24** | 81.58 | 81.58 | 77.63 |
| abstract-algebra | 60.00 | 59.00 | **62.00** | 60.00 |
| nutrition | **81.00** | 78.50 | 80.50 | 79.50 |
| high-school-biology | **80.50** | 78.50 | 79.50 | 76.00 |
| formal-fallacies | 48.50 | **50.50** | 49.00 | 49.00 |
| tracking-3 | 50.00 | 46.00 | 47.50 | **51.50** |
| Average | 71.13 | 70.20 | **71.16** | 70.63 |

Table 15: Performance comparison between BSR and SUR[D,BSR] with DeBERTa as the surrogate model with ($k = 3$ shots) for Llama2-13b-chat.

| Dataset | BSR | SUR[D,BSR] |
|---|---|---|
| medical-genetics | 87.00 | 85.00 |
| prof-psychology | 70.00 | 72.50 |
| formal-logic | 59.52 | 59.52 |
| moral-disputes | 78.00 | 84.50 |
| public-relations | 79.09 | 76.36 |
| comp-security | 76.00 | 77.00 |
| astronomy | 80.26 | 80.26 |
| abstract-algebra | 58.00 | 59.00 |
| nutrition | 77.5 | 79.00 |
| high-school-biology | 76.50 | 79.00 |
| formal-fallacies | 52.00 | 45.5 |
| tracking-3 | 44.00 | 46.50 |
| Average | 69.82 | **70.30** |

Table 16: Performance comparison between BSR, SUR[D,BSR], and **PRE[D,BSR]** with ($k = 3$ shots) for Qwen2.5-3b.

| Dataset | BSR | SUR[D,BSR] | PRE[D,BSR] |
|---|---|---|---|
| medical-genetics | 88 | 90 | 87 |
| prof-psychology | 80 | 82.5 | 81.5 |
| formal-logic | 71.43 | 66.67 | 68.25 |
| moral-disputes | 83.5 | 80 | 82.5 |
| public-relations | 79.09 | 79.09 | 80.91 |
| comp-security | 84 | 87 | 85 |
| astronomy | 85.53 | 88.16 | 88.82 |
| abstract-algebra | 64 | 63 | 63 |
| nutrition | 84.5 | 85.5 | 85 |
| high-school-biology | 87.5 | 87 | 89 |
| formal-fallacies | 49 | 50 | 50.5 |
| tracking-3 | 48 | 48 | 49 |
| Average | 75.38 | 75.57 | **75.87** |

## C.6 RESULTS ON LLAMA-3-70B

Here's an explanation for the Llama-3-70B results following the style of your previous explanation:

Here we present results on Llama-3-70B for $k = 3$ shots on the Mixture of Tasks setting. We compare both BSR and Cosine similarity baselines with our IF-based methods: $\text{SUR}_{[D,BSR]}$, $\text{PRE}_{[D,BSR]}$, $\text{SUR}_{[D,Cos]}$, and $\text{PRE}_{[D,Cos]}$. We see in Table 17 that for a large-scale model like Llama-3-70B, the pretrained gradients combined with cosine similarity ($\text{PRE}_{[D,Cos]}$) achieve the highest average performance at 83.256%. Notably, the pretrained-gradient based methods outperform the surrogate-model based methods in a pattern similar to the Qwen2.5-3b results. This suggests that the pretrained gradients in the newer-larger models are more representative of model's inductive bias than the older-smaller models.

Table 17: Performance comparison between BSR, $\text{SUR}_{[D,BSR]}$, $\text{PRE}_{[D,BSR]}$, Cos, $\text{SUR}_{[D,Cos]}$, and $\text{PRE}_{[D,Cos]}$ for Llama-3-70B.

| Dataset | BSR | $\text{SUR}_{[D,BSR]}$ | $\text{PRE}_{[D,BSR]}$ | Cos | $\text{SUR}_{[D,Cos]}$ | $\text{PRE}_{[D,Cos]}$ |
|---|---|---|---|---|---|---|
| medical-genetics | 95 | 94 | 97 | 97 | 98 | 98 |
| prof-psychology | 92.5 | 91 | 90.5 | 91.5 | 91.5 | 90.5 |
| formal-logic | 72.22 | 77.78 | 74.6 | 73.81 | 73.02 | 79.37 |
| moral-disputes | 89 | 91 | 90 | 87 | 90.5 | 91 |
| public-relations | 86.36 | 84.55 | 84.55 | 83.64 | 85.45 | 86.36 |
| comp-security | 86 | 86 | 86 | 87 | 88 | 87 |
| astronomy | 98.68 | 98.68 | 97.37 | 98.03 | 97.37 | 99.34 |
| abstract-algebra | 68 | 66 | 67 | 65 | 65 | 66 |
| nutrition | 95.5 | 93.5 | 93.5 | 95.5 | 95.5 | 94 |
| high-school-biology | 93.5 | 94 | 93.5 | 94 | 94.5 | 94 |
| formal-fallacies | 59 | 66.5 | 60.5 | 59.5 | 63 | 66 |
| tracking-3 | 43.5 | 46 | 47 | 48 | 45.5 | 47.5 |
| **Average** | 81.605 | 82.418 | 81.793 | 81.665 | 82.278 | **83.256** |

## C.7 ABLATION ANALYSIS ON THE BENEFITS OF COMBINING IF WITH SEMANTIC SIMILARITY METHODS

We conduct an ablation analysis to examine the benefit of using BertScore-Recall and Cosine Similarity in conjunction with our IF-based methods. We want to quantify how effective these methods are at selecting the same training task demonstration as the test task in the 3 shots selected (e.g if the test task is from MMLU-abstract-algebra, then the demonstration retrieved is also from the MMLU-abstract-algebra train set in our demonstration pool). We find that BertScore-Recall selects the same task **33.90%** of the time and Cosine Similarity selects the same task **36.03%** of the time. In our MoT approach, we initially retrieve the top $2k$ candidate demonstrations using similarity-based methods, and subsequently re-rank and select the final $k$ examples based on IF scores. This staged retrieval process effectively narrows the pool to semantically relevant demonstrations, enabling the IF-based re-ranking to more precisely exploit the model's inductive biases.

## C.8 QUALITATIVE ANALYSIS

For MoT, to understand the merits of our method, we compare demonstrations selected via BSR and $\text{SUR}_{[D,BSR]}$ on the MMLU-abstract-algebra dataset:

**Example 1:**

**Q:** Compute the product in the given ring. $(2, 3)(3, 5)$ in $\mathbb{Z}_5 \times \mathbb{Z}_9$

**Options:** (B) (3,1)    (C) (1,6)

**BSR Shots.**

1. **Q:** Statement 1 | Every element of a group generates a cyclic subgroup of the group. Statement 2 | The symmetric group $S_{10}$ has 10 elements. **Options:** (A) True, True    (C) True, False **Answer:** (C)

2. **Q:** Statement 1 | Every function from a finite set onto itself must be one-to-one. Statement 2 | Every subgroup of an abelian group is abelian. **Options:** (A) True, True    (D) False, True **Answer:** (A)

3. **Q:** How many attempts should you make to cannulate a patient before passing the job on to a senior colleague, according to the medical knowledge of 2020? **Options:** (A) 4    (B) 3    (C) 2    (D) 1 **Answer:** (C)

$\text{SUR}_{[\text{D,BSR}]}$.

1. **Q:** Statement 1 | Every function from a finite set onto itself must be one-to-one. Statement 2 | Every subgroup of an abelian group is abelian. **Options:** (A) True, True    (D) False, True **Answer:** (A)

2. **Q:** Olivia used the rule "Add 11" to create the number pattern shown below: 10, 21, 32, 43, 54. Which statement about the number pattern is true? **Options:** (B) The number pattern will never have two even numbers next to each other. (D) If the number pattern started with an odd number, then the pattern would have only odd numbers in it. **Answer:** (B)

3. **Q:** Tomorrow is 11/12/2019. What is the date one year ago from today in MM/DD/YYYY format? **Options:** (B) 11/11/2018    (C) 08/25/2018 **Answer:** (B)

We can see that while BSR selects more semantically relevant samples, $\text{SUR}_{[\text{D,BSR}]}$'s selected shots guide the model toward the correct answer (C) instead of (B) by encouraging more structured reasoning.

**Example 2:**

**Q:** Statement 1 | If $R$ is an integral domain, then $R[x]$ is an integral domain. Statement 2 | If $R$ is a ring and $f(x)$ and $g(x)$ are in $R[x]$, then

$$\deg(f(x)g(x)) = \deg f(x) + \deg g(x).$$

**Options:** (C) True, False    (B) False, False

**BSR Shots.**

1. **Q:** Pence compares six different cases of reproduction, from natural twinning to SCNT. What conclusion does he draw from this comparison? **Options:** (A) SCNT is not a different kind of reproduction because there are no morally relevant differences between it and other permissible means of reproduction. (B) Because there is a low risk of harm for natural twinning, there will be a low risk of harm for SCNT. (C) Both A and B (D) Neither A nor B **Answer:** (A)

2. **Q:** Statement 1 | Every element of a group generates a cyclic subgroup of the group. Statement 2 | The symmetric group $S_{10}$ has 10 elements. **Options:** (A) True, True    (C) True, False **Answer:** (C)

3. **Q:** Statement 1 | Every function from a finite set onto itself must be one-to-one. Statement 2 | Every subgroup of an abelian group is abelian. **Options:** (A) True, True    (D) False, True **Answer:** (A)

$\text{SUR}_{[\text{D,BSR}]}$.

1. **Q:** Statement 1 | Every function from a finite set onto itself must be one-to-one. Statement 2 | Every subgroup of an abelian group is abelian. **Options:** (A) True, True    (D) False, True **Answer:** (A)

2. **Q:** Select the best translation into predicate logic. George borrows Hector's lawnmower. ($g$: George; $h$: Hector; $l$: Hector's lawnmower; $Bxyx$: $x$ borrows $y$ from $z$). **Options:** (A) $B_{lgh}$ (B) $B_{hlg}$ (C) $B_{glh}$ (D) $B_{ghl}$ **Answer:** (C)

3. **Q:** Statement 1 | Every element of a group generates a cyclic subgroup of the group. Statement 2 | The symmetric group $S_{10}$ has 10 elements. **Options:** (A) True, True    (C) True, False **Answer:** (C)

Here again, BSR selects the more semantically relevant shots (with the top three shots ordered in ascending order of relevance), while $\text{SUR}_{[D,BSR]}$ selects less semantically similar but more influential shots, which ultimately improves model performance.

# D  EXTENDED NOISY ICL RESULTS

## D.1  VARYING NOISE LEVELS

We also test whether our method can perform well on varying noise levels in the dataset. To test this, we create 2 datasets of MRPC with 10% and 30% noise added. As seen in Table 18, in both the cases IF Averaging outperformed other baselines. With the DataInf configuration performing better for the 10% noise dataset and the LiSSA configuration performing better for the 30% noise dataset.

Table 18: MRPC 10% and 30% Noise added results using various methods (top 2 performers in bold).

| | Method | MRPC 0.1 | MRPC 0.3 |
|---|---|---|---|
| | **RAND** | 70.1 | 70.8 |
| | **BSR** | 70.3 | 70.3 |
| | **COS** | 70.6 | 70.6 |
| | **BM25** | 73.5 | 73.0 |
| PRU-0.1 | RAND$_{[Cos]}$ | 68.6 | 69.9 |
| | SUR$_{[D,Cos]}$ | 68.4 | 69.9 |
| | SUR$_{[L,Cos]}$ | 68.9 | 68.9 |
| | RAND$_{[BSR]}$ | 69.1 | 68.6 |
| | SUR$_{[D,BSR]}$ | 66.4 | 69.1 |
| | SUR$_{[L,BSR]}$ | 66.7 | 71.8 |
| AVG-0.5 | SUR$_{[D,Cos]}$ | **75.0** | 70.3 |
| | SUR$_{[L,Cos]}$ | 69.9 | **76.0** |
| | SUR$_{[D,BSR]}$ | **73.8** | 70.8 |
| | SUR$_{[L,Cos]}$ | 72.1 | **75.7** |

## D.2  VARYING HYPERPARAMETERS

Here we provide results for different IF pruning and IF averaging hyperparameters that we tested with varying levels of noise in Table 19 and Table 20.

## D.3  EFFECTIVENESS OF IFS IN IDENTIFYING MISLABELED DATA

We conduct a toy experiment to evaluate the effectiveness of IF-based methods in detecting noisy samples. We introduce 20% noise to the datasets and compute IF values using the Surrogate Model approach. We then calculate the percentage of noisy samples in the top 100 values selected by our IF methods. Results are presented in Table 21

As shown in the table, IF-based methods are highly effective in identifying mislabeled data, significantly aiding demonstration selection in Noisy ICL.

# E  MEMORY CONSUMPTION

To analyze the added computational costs associated with IFs, we calculate the maximum GPU memory consumption while performing demonstration selection with the Pretrained Gradients-DataInf $\text{PRE}_D$ and Surrogate Model-DataInf $\text{SUR}_D$ methods. The experiments are performed on 4 NVIDIA RTX 6000 Ada Generation GPUs. The maximum memory consumption for Pre-D was 18,188 MiB, while for Sur-D it was 7,998 MiB. These memory requirements are relatively modest, and the use of IFs can be justified given the benefits they provide.

Table 19: MRPC results using various methods and configurations for 10% and 30% Noise.

| | Method | MRPC 0.1 | MRPC 0.3 |
|---|---|---|---|
| **PRU-0.2** | $\text{RAND}_{[\text{Cos}]}$ | 68.9 | 69.1 |
| | $\text{SUR}_{[\text{D,Cos}]}$ | 71.1 | 69.9 |
| | $\text{SUR}_{[\text{L,Cos}]}$ | 70.6 | 72.3 |
| | $\text{RAND}_{[\text{BSR}]}$ | 70.1 | 68.1 |
| | $\text{SUR}_{[\text{D,BSR}]}$ | 70.1 | 70.1 |
| | $\text{SUR}_{[\text{L,BSR}]}$ | 71.3 | 66.7 |
| **PRU-0.3** | $\text{RAND}_{[\text{Cos}]}$ | 69.9 | 69.6 |
| | $\text{SUR}_{[\text{D,Cos}]}$ | 71.8 | 69.4 |
| | $\text{SUR}_{[\text{L,Cos}]}$ | 71.3 | 71.3 |
| | $\text{RAND}_{[\text{BSR}]}$ | 69.4 | 67.9 |
| | $\text{SUR}_{[\text{D,BSR}]}$ | 70.3 | 71.1 |
| | $\text{SUR}_{[\text{L,BSR}]}$ | 72.1 | 73.0 |
| **AVG-0.4** | $\text{SUR}_{[\text{D,Cos}]}$ | 71.8 | 69.9 |
| | $\text{SUR}_{[\text{L,Cos}]}$ | 69.4 | 73.0 |
| | $\text{SUR}_{[\text{D,BSR}]}$ | 72.1 | 75.4 |
| | $\text{SUR}_{[\text{L,BSR}]}$ | 67.4 | 70.3 |
| **AVG-0.6** | $\text{SUR}_{[\text{D,Cos}]}$ | 70.3 | 73.5 |
| | $\text{SUR}_{[\text{L,Cos}]}$ | 74.3 | 73.3 |
| | $\text{SUR}_{[\text{D,BSR}]}$ | 70.3 | 71.8 |
| | $\text{SUR}_{[\text{L,BSR}]}$ | 71.3 | 73.8 |

Table 20: Noisy ICL Accuracy with different hyper-parameters for our methods.

| | Method | MRPC 0.2 | QNLI 0.2 | SST2 0.2 | QQP 0.2 |
|---|---|---|---|---|---|
| **PRU-0.2** | $\text{RAND}_{[\text{Cos}]}$ | 68.1 | 68.6 | 86.6 | 70.0 |
| | $\text{SUR}_{[\text{D,Cos}]}$ | 67.7 | 67.2 | 86.2 | 70.0 |
| | $\text{SUR}_{[\text{L,Cos}]}$ | 71.3 | 65.8 | 86.8 | 67.2 |
| | $\text{RAND}_{[\text{BSR}]}$ | 69.1 | 72.2 | 85.4 | 69.8 |
| | $\text{SUR}_{[\text{D,BSR}]}$ | 70.3 | 67.4 | 81.8 | 71.6 |
| | $\text{SUR}_{[\text{L,BSR}]}$ | 70.1 | 66.4 | 85.2 | 70.8 |
| **PRU-0.3** | $\text{RAND}_{[\text{Cos}]}$ | 70.1 | 68.4 | 87.0 | 67.4 |
| | $\text{SUR}_{[\text{D,Cos}]}$ | 68.8 | 69.2 | 84.6 | 70.6 |
| | $\text{SUR}_{[\text{L,Cos}]}$ | 70.6 | 68.2 | 86.6 | 72.6 |
| | $\text{RAND}_{[\text{BSR}]}$ | 70.8 | 65.8 | 86.0 | 70.8 |
| | $\text{SUR}_{[\text{D,BSR}]}$ | 70.6 | 67.4 | 84.8 | 72.0 |
| | $\text{SUR}_{[\text{L,BSR}]}$ | 69.4 | 68.2 | 87.4 | 68.4 |
| **AVG-0.4** | $\text{SUR}_{[\text{D,Cos}]}$ | 75.7 | 71.4 | 91.6 | 62.6 |
| | $\text{SUR}_{[\text{L,Cos}]}$ | 73.3 | 67.2 | 90.6 | 75.2 |
| | $\text{SUR}_{[\text{D,BSR}]}$ | 74.3 | 68.8 | 89.8 | 65.2 |
| | $\text{SUR}_{[\text{L,BSR}]}$ | 56.6 | 72.8 | 78.0 | 73.0 |
| **AVG-0.6** | $\text{SUR}_{[\text{D,Cos}]}$ | 73.5 | 69.2 | 87.2 | 66.2 |
| | $\text{SUR}_{[\text{L,Cos}]}$ | 72.1 | 68.6 | 84.6 | 70.8 |
| | $\text{SUR}_{[\text{D,BSR}]}$ | 73.5 | 69.2 | 94.2 | 71.8 |
| | $\text{SUR}_{[\text{L,BSR}]}$ | 72.3 | 65.8 | 94.2 | 75.4 |

# F   SCALABILITY TO LARGE MODELS

We compare the time it takes to extract test set gradients and compute influence scores. We compute IF via the DataInf method, comparing RoBERTa-Large (125 million parameters), Llama2-13b-chat (13 billion parameters), and Llama2-70b-chat (70 billion parameters) on the MMLU-moral-disputes dataset with 200 test samples. The results are provided in Table 22.

Table 21: Percentage of noisy samples in the top 100 values selected by IF methods.

| Dataset | DataInf | LiSSA |
|---------|---------|-------|
| MRPC    | 83%     | 66%   |
| QNLI    | 54%     | 86%   |
| QQP     | 79%     | 95%   |
| SST-2   | 90%     | 96%   |

Table 22: Time taken for extracting test gradients and computing IF across different models.

| Model            | Test Gradients (s) | Computing IF (s) |
|------------------|--------------------|------------------|
| RoBERTa          | 7.447              | 35.72            |
| Llama-2-13b-chat | 68.5               | 4.69             |
| Llama-2-70b-chat | 257.64             | 8.81             |

The relationship between model size and inference time grows sublinearly, with time increasing at roughly the square root of the model size. We also see that it takes longer to compute the IF in the RoBERTa model due to the fine-tuning process[4].

Additionally, the time required to compute IF using the TracIn method on Llama-2-13b-chat is just $3.576 \times 10^{-6}$ seconds. This highlights the significant speed advantage offered by TracIn.

We would like to emphasize that practitioners have the flexibility to choose between our models and methods based on their specific needs. If computational efficiency is the priority, the significantly faster surrogate model approach can be used. Conversely, if high accuracy is desired and compute is not a concern, a fine-tuned LLM is a better alternative.

# G  SCALABILITY TO LARGE DATASETS

For larger datasets, we compare the time taken to extract test gradients and compute IF for 100 samples, 200 samples, and 1000 samples in Table 23.

Table 23: Time taken for computing test gradients and influence functions (IF) across different models and sample sizes.

| Model & Samples                 | Test Gradients (s) | Computing IF (s) |
|---------------------------------|--------------------|------------------|
| RoBERTa (100 Samples)           | 6.7                | 26.4             |
| RoBERTa (200 Samples)           | 7.4                | 35.7             |
| RoBERTa (1000 Samples)          | 67.8               | 273.7            |
| Llama-2-13b-chat (100 Samples)  | 41.0               | 3.6              |
| Llama-2-13b-chat (200 Samples)  | 68.5               | 4.7              |
| Llama-2-13b-chat (1000 Samples) | 410.8              | 35.4             |

A 10x increase in sample size corresponds to an approximately 10x increase in computational time, indicating a linear relationship between sample size and computational time.

Finally, in MoT, the computational time of IF can further be optimized by only computing IF for the $2k$ shots being pruned by BSR or Cosine similarity instead of the entire set of training demonstrations. Another optimization to the DataInf code could be replacing their handling of gradients with tensor operations instead of the current dict of dicts format. This enables the use of GPU processing for influence computation instead of CPU and can offer a considerable runtime speedup.

---

[4]We fine-tuned the RoBERTa model and computed IF using gradients from its LoRA-adapted components. In contrast, no fine-tuning was performed on the larger LLaMA2-13B-Chat model; consequently, only Layer-Norm weights produced non-zero gradients. This significantly reduced the parameter space and computational cost of IF estimation, as IFs depend on the inverse Hessian with respect to model parameters.

## H IF-BASED DEMONSTRATIONS AS A BACKDOOR DEFENSE STRATEGY TO MITIGATE BACKDOOR ATTACKS

To evaluate the robustness of our influence-based demonstration selection method under a different class of adversarial noise, we extend it to a **task-agnostic** backdoor defense setting. This scenario reflects practical constraints where task-specific labeled data may be unavailable at inference time, yet a broader pool of related examples is accessible.

We use our $\textbf{SUR}_{\textbf{[D,BSR]}}$ strategy to the SST-2 dataset (Wang, 2018), which has been poisoned with two distinct forms of backdoor attacks: (1) *AddSent* (Dai et al., 2019), where a semantic trigger is introduced by inserting an innocuous sentence (e.g., "I watched this 3D movie last weekend"), and (2) *Style* (Qi et al., 2021), a style-based attack that performs backdoor poisoning through text style transfer (e.g., transforming text into a Biblical style).

We evaluate the *Attack Success Rate (ASR)*, defined as the percentage of non-target label test instances that are misclassified as the target label when evaluated on a poisoned dataset. For comparison, we consider a task-aware demonstration selection baseline (Diao et al., 2023), which selects 5 demonstrations from a clean, task-specific dataset. All experiments are conducted using the LLaMA3-8B model (Grattafiori et al., 2024). We present our results in Table 24.

| SST-2 | AddSent | Style |
|---|---|---|
| | **ASR** | |
| No Defense | 100.00 | 98.68 |
| Task-Aware | 69.41 | 32.02 |
| $\text{SUR}_{[D,BSR]}$ | 58.22 | 10.86 |

Table 24: Defense results on AddSent and Style backdoor attacks. ASR: Attack Success Rate (the lower the better).

Our results show that our IF-based indirect-ICL paradigm can effectively mitigate various types of backdoor attacks. This reveals that even without task-specific data, our results demonstrate that the IF-based indirect-ICL paradigm effectively mitigates a range of backdoor attacks, showing a 32.89% reduction in ASR, on average. Notably, even in the absence of task-specific data, demonstrations selected based on a model's inductive bias can provide a strong defense mechanism, highlighting the potential of task-agnostic strategies as an effective defense mechanism against backdoor attacks.

