# OpenReview forum: "Unraveling Indirect In-Context Learning Using Influence Functions"
_ICLR.cc/2026/Conference — Submitted to ICLR 2026_

### Official Review · Reviewer_zbgC · 2025-10-26

**Soundness:** 1
**Presentation:** 2
**Contribution:** 2
**Rating:** 2
**Confidence:** 3

**Summary:**

This paper introduces a new paradigm for generalized in-context learning, called indirect in-context learning (Indirect ICL), which addresses scenarios where the standard assumption of having a clean, task-aligned demonstration pool is violated. The paper considers two main tasks to demonstrate the effectiveness of the proposed paradigm: 1) Mixture of Tasks (MoT), where the pool contains demonstrations for many tasks, among which only a fraction is relevant to the test tasks, 2) Noisy ICL, where the pool contains corrupted demonstrations.

The paper proposes using influence functions (IFs) to capture the "inductive bias" of task demonstrations with the end-task. There are some empirical results showing the superiority of the proposed method compared to a few constructed baseline methods.

**Strengths:**

- The paper studies an interesting and important problem of task demonstration selection for ICL, including task misalignment and noisy or corrupted task demonstrations in the pool.
- The paper offers some interesting insights via experiments, e.g., that pruning hurts rather than helps performance and that DataInf is better than TracIn as an IF method.

**Weaknesses:**

- The proposed method requires training the model to obtain the IF of the demonstrations. As such, a fair experimental comparison should involve training-based baseline methods, for example, SFT on the ICL data.
- Apart from that, the experiments are more like an ablation of the proposed method, missing performance comparison with baseline methods, including several ICL selection methods, for example, [1,2]
- Missing analysis: It is quite counterintuitive to grasp why a surrogate model outperforms running IF on the LLM itself. The authors should explain more about this point.
- Missing (actual) computational cost comparison: On top of that, what is the speedup gained by utilizing the surrogate model? How is the speed compared to other training-free baselines? While I appreciate that the paper includes a complexity analysis, it does not reflect the real computation incurred during deployment.
- Question on novelty: the idea of using IF for ICL has been explored previously, for example, in [1,3]. Notably, [1] constructs a "surrogate model" in the form of a linear regression on the hidden states of the LLM itself and can compute the IF of demonstrations quickly. Given the similarity, I hope the authors can provide an explanation comparing the proposed method and [1,3], and also include some experiments comparing the methods.



[1] Zhou et al., DETAIL: Task Demonstration Attribution for Interpretable In-context Learning, NeurIPS 2024.

[2] Peng et al., Revisiting Demonstration Selection Strategies in In-Context Learning, EMNLP 2024.

[3] M.S. et al., In-Context Learning Demonstration Selection via Influence Analysis, arXiv 2402.11750.

**Questions:**

- What is a precise definition of indirect in-context learning? I'm looking for either a definitive language description or a math formulation. I can't seem to find it in the paper.

The other questions are in the weaknesses.

---

> ### Author Response · Authors · 2025-11-23
> **Rebuttal to Review [1/3]**
>
> We are grateful to the reviewer for their feedback of our work, and are appreciative of their time, effort, and consideration. Below we provide more discussion on the points raised.
>
> ---
>
> * **Comparison with another training based baseline method:**
>
>     We agree with the reviewers suggestion and perform experiments using MoICL [1] a recent ACL 2025 paper that treats subsets of demonstrations as experts and learns a weighting function to merge their output distributions based on a training set via gradient-based optimization. We compare the results on Llama-2-13b-chat with 3 demonstrations.
>
>     | Dataset                 | MOICL | SUR[D,BSR] |
>     |:------------------------|------:|-----------:|
>     | medical-genetics        | 77.00 |      86.00 |
>     | prof-psychology         | 70.00 |      72.00 |
>     | formal-logic            | 57.14 |      57.14 |
>     | moral-disputes          | 79.50 |      80.50 |
>     | public-relations        | 68.18 |      79.09 |
>     | comp-security           | 75.00 |      76.00 |
>     | astronomy               | 77.63 |      80.26 |
>     | abstract-algebra        | 53.00 |      72.00 |
>     | nutrition               | 75.50 |      79.50 |
>     | high-school-biology     | 75.00 |      80.50 |
>     | formal-fallacies        | 46.00 |      47.50 |
>     | tracking-3              | 27.00 |      43.50 |
>     | **Average**             | 65.07 | **71.16** |
>
>     As we can clearly see, our method vastly over performs another gradient based training baseline.
>
> ---
>
> * **Experiments look like ablations and missing experiments:**
>
>     We appreciate the reviewer's concern regarding the breadth of our baseline comparisons. However, we would like to clarify that our baseline setup is consistent with those used in prior published work on demonstration selection. For example, the reviewer cited the ACL 2024 paper by Peng et al. "Revisiting Demonstration Selection Strategies in In-Context Learning," which primarily compared against four baselines, the same as ours. Additionally, another paper from ACL 2024 on ICL selection, He et al., 2024; "Using Natural Language Explanations to Improve Robustness of In-Context Learning" uses a similar set of baselines.
>
>     That said, we acknowledge the reviewer's suggestion and are happy to incorporate results on the aforementioned ICL based baselines. We compare our method with DETAIL [2], InfICL [3], and ConE [4] on Llama2-13b-chat with 3 demonstrations. For DETAIL, we use their implementation provided in their codebase. InfICL did not come with a code release so we borrowed DETAIL's implementation of their algorithm and added a vectorized LiSSA computation for higher efficiency. For ConE we use their code release with the default parameters.
>
>     | Dataset              | DETAIL | InfICL | ConE  | SUR[D,BSR] |
>     |:---------------------|-------:|-------:|------:|-----------:|
>     | medical-genetics     |  77.00 |  82.00 | 81.00 |      86.00 |
>     | prof-psychology      |  66.50 |  68.00 | 71.00 |      72.00 |
>     | formal-logic         |  61.90 |  65.08 | 61.90 |      57.14 |
>     | moral-disputes       |  79.50 |  79.50 | 81.50 |      80.50 |
>     | public-relations     |  70.00 |  75.45 | 75.45 |      79.09 |
>     | comp-security        |  71.00 |  78.00 | 80.00 |      76.00 |
>     | astronomy            |  77.63 |  78.29 | 80.92 |      80.26 |
>     | abstract-algebra     |  71.00 |  54.00 | 62.00 |      72.00 |
>     | nutrition            |  76.00 |  79.50 | 80.00 |      79.50 |
>     | high-school-biology  |  75.50 |  78.50 | 76.50 |      80.50 |
>     | formal-fallacies     |  48.00 |  46.00 | 47.50 |      47.50 |
>     | tracking-3           |  34.00 |  36.50 | 48.00 |      43.50 |
>     | **Average**          | 67.34 | 68.40 | 70.48 | **71.16** |
>
>     As can be observed, our method still outperforms other IF based methods and ConE as well, indicating the effectiveness of our method for demonstration selection in Indirect ICL.
>
> ---

---

> > ### Author Response · Authors · 2025-11-23
> > **Rebuttal to Review [2/3]**
> >
> > * **Why Surrogate Model results outperform Pretrained Model results:**
> >
> >     We agree that it may seem counterintuitive that a smaller surrogate model can outperform running IF directly on the LLM. However, the surrogate is fine-tuned on the same candidate demonstrations and validation tasks, so its gradients are better aligned with the downstream objective, leading to more informative influence scores; in contrast, the non-finetuned LLM's gradients remain optimized for a broad language-modeling objective and can be less task-specific. This behavior is consistent with prior influence-based data selection work, which shows that relatively small, task-aligned proxy models can be highly effective for influence estimation [1,2]. Empirically, our Mixture-of-Tasks results (Sec. 4.3) show surrogate-based IF methods performing best for medium sized models, while results on newer/larger LLMs in Appendix. C.5–C.6) show pretrained-gradient variants becoming competitive or superior indicating that the quality of the pretrained gradients also plays a role in the informativeness of the IF based selection from its gradients.
> >
> > ---
> >
> > * **Actual computation cost numbers:**
> >
> >     We thank the reviewer for raising this point. Our paper does include empirical computational analyses beyond asymptotic complexity specifically, Appendix E (GPU memory usage), Appendix F (scalability to larger LLMs), and Appendix G (scalability to larger datasets). These experiments directly measure the actual cost incurred during selection.
> >
> >     The results show that surrogate-model IF methods consistently require substantially less GPU memory and lower per-example compute cost compared to computing gradients on the full LLM, since influence computation on RoBERTa-based surrogates involves far fewer parameters. This aligns with the observed efficiency trends in Appendix E–G, where surrogate methods scale more smoothly with model size and demonstration pool size than the pretrained-gradient IF variants.
> >
> >     Importantly, the surrogate model’s fine-tuning cost is a one-time offline step, after which inference requires only forward and backward passes on the small surrogate model. As a result, the deployment-time cost is lightweight and sits between the very fast training-free baselines (COS, BM25) and the more expensive LLM-gradient IF methods. We will revise the article to make these empirical findings more prominent and explicitly summarize the implications for deployment-time.
> >
> > ---
> >
> > * **Novelty in method and problem formulation:**
> >
> >     We thank the reviewer for the question. While InfICL and DETAIL both apply influence functions in an ICL context, our method differs substantially in methodology, mathematical formulation, and problem setting. Methodologically, InfICL trains a small 3-layer classifier on LLM embeddings and computes IF scores with respect to this classifier’s parameters; in contrast, we fine-tune a transformer-based surrogate model (RoBERTa) and compute IF using its gradients or use pretrained LLM gradients directly, yielding a much richer gradient geometry. DETAIL, on the other hand, focuses on demonstration attribution, not selection, and formulates a closed-form kernel ridge regression over hidden states. Their surrogate is not trained and does not approximate model gradients; mathematically it is fundamentally different from our gradient–Hessian–vector formulation.
> >
> >     More importantly, the problem setting we address is different. Both InfICL and DETAIL assume standard ICL where demonstrations come from the same task. Our work targets Indirect ICL, which includes (1) Mixture-of-Tasks (MoT), where most demonstrations are from unrelated tasks, and (2) Noisy ICL, where demonstrations may be mislabeled or adversarially corrupted. DETAIL shows how to detect noisy demonstrations, but not how to select demonstrations under systematic noise. Our contributions include two principled noisy-ICL selection strategies (IF Pruning and IF Averaging) and extensive evaluation under adversarial corruption (Appendix H). Crucially, neither prior work addresses MoT at all, whereas our two-stage selection procedure is specifically designed for task-mismatched pools an increasingly common real-world scenario. Finally, DETAIL was published much after InfICL was released. A similar argument against its "novelty" can be made since they both are using IFs to perform ICL, but we believe that our methodological and problem formulation separate us from them enough to claim scientific novelty. Finally, we have performed experiments comparing these methods with ours showing our superiority over them in Indirect ICL.

---

> > > ### Author Response · Authors · 2025-11-23
> > > **Rebuttal to Review [3/3]**
> > >
> > > * **Precise Definition of Indirect-Incontext Learning**
> > >
> > >    Thank you for this comment; we provide both a natural language and a mathematical definition of Indirect ICL and will definitely add both to a revised version of the paper.
> > >
> > >    The following is the natural language formulation:
> > >
> > >        Indirect ICL is the problem of selecting in-context demonstrations from a task-agnostic or noisy candidate pool, where many examples may not belong to the end task or may contain label corruption, while still aiming to maximize test-time performance when the LLM is prompted.
> > >
> > >    We also provide a mathematical formulation below which we will add to the paper in proper mathematical symbols:
> > >
> > >
> > >        Indirect ICL considers a candidate pool:
> > >        D = {(x_i, y_i, t_i)} for i=1...N
> > >        where the task identities t_i are unknown and may not match the test task t_test, and some labels y_i may also be noisy or corrupted.
> > >
> > >        The goal is to select a demonstration set:
> > >        S⊂D, |S|=k
> > >        from this task-agnostic or noisy pool and then predict the test label using the prompt T(S, x_test).
> > >
> > >        The objective is:
> > >        S* = argmax over S⊂D, |S|=k  of P(y_test|T(S, x_test))
> > >        without assuming that the pool matches the test task or that labels are clean.
> > >
> > >     We will add this precise formulation to the updated version of our manuscript.
> > >
> > > ---
> > >
> > > Thank you again for taking the time to review our paper. We hope that we addressed some of your concerns enough to warrant an increase in score.
> > >
> > > ---
> > >
> > > References:
> > >
> > > [1] Hong, Giwon, et al. "Mixtures of in-context learners." Proceedings of the 63rd Annual Meeting of the Association for Computational Linguistics (Volume 1: Long Papers). 2025.
> > >
> > > [2] Zhou et al., DETAIL: Task Demonstration Attribution for Interpretable In-context Learning, NeurIPS 2024.
> > >
> > > [3] M.S. et al., In-Context Learning Demonstration Selection via Influence Analysis, arXiv 2402.11750.
> > >
> > > [4] Peng et al., Revisiting Demonstration Selection Strategies in In-Context Learning, EMNLP 2024.

---

> > > > ### Author Response · Authors · 2025-11-28
> > > > **Happy to engage in further discussion**
> > > >
> > > > Thank you again for the thoughtful review. We hope our response addressed your concerns and are happy to discuss if you have any further questions!

---

### Official Review · Reviewer_zjg3 · 2025-10-31

**Soundness:** 2
**Presentation:** 3
**Contribution:** 2
**Rating:** 4
**Confidence:** 4

**Summary:**

This paper introduces a generalized paradigm for In-Context Learning (ICL) termed Indirect In-Context Learning, which addresses demonstration selection for two challenging, real-world scenarios: a Mixture of Tasks setting and a Noisy ICL setting. The authors propose using Influence Functions (IFs) as a tool to select informative demonstrations from a pool that is either task-heterogeneous or contains label noise. They demonstrate that combining IFs from a surrogate model with traditional semantic similarity metrics like BERTScore-Recall can lead to modest performance gains in the MoT setting. For the noisy setting, they show that IF-based pruning or averaging can improve accuracy on mislabeled data and reduce the success rate of backdoor attacks.

**Strengths:**

- Originality in Problem Formulation: The paper identifies and formalizes a relevant and under-explored problem—ICL when direct, clean task demonstrations are unavailable. The MoT and Noisy ICL settings are practical and important.

- Thorough Empirical Evaluation: The authors conduct extensive experiments across multiple LLMs, datasets, and settings (MoT, noisy, adversarial), which adds credibility to their findings.

- Clarity and Scope: The paper is clearly written, and the exploration of both surrogate-model and pretrained-gradient approaches to IFs is a valuable comparison.

- Actionable Insights: The finding that a simple combination of semantic similarity and IF-based re-ranking is effective is a practical takeaway.

**Weaknesses:**

- Limited Empirical Improvement: The performance gains are consistently small, often within a few percentage points, and their statistical significance is not established. This undermines the claim that IFs provide a substantial benefit.

- Incremental Technical Contribution: The methodology is an application of existing tools rather than a novel algorithmic or theoretical contribution. The two-stage selection process is a simple ensemble of existing techniques.

- Scalability and Cost Concerns: While computational complexity is analyzed, the practical cost of fine-tuning surrogate models or computing gradients from very large models (e.g., Llama-3-70B) for every task is non-trivial and not sufficiently justified by the modest gains achieved.

- Lack of Ablation on IF Variants: The paper tests multiple IF methods but provides limited analysis explaining why certain methods (e.g., DataInf) outperform others (e.g., TracIn) beyond the lack of second-order information.

**Questions:**

Please refer to Weakness.

---

> ### Author Response · Authors · 2025-11-23
> **Rebuttal to Review [1/3]**
>
> We are grateful to the reviewer for their detailed feedback of our work, and are appreciative of their time, efforts, and consideration. Below we provide more discussion on some of the points raised.
>
> ---
>
> * **Limited Empirical Improvements and Statistical significance:** While the absolute performance gains may appear modest, 0.37\% for 3-shot MoT, 1.45\% for 5-shot MoT, and 2.94\% for Noisy ICL, they are achieved under highly constrained conditions: only three to five in-context demonstrations, sampled from pools largely unrelated to the target task. In this challenging setting, even the strongest baselines outperform random selection by just 1.41\%, highlighting the difficulty of the task. Notably, our approach outperforms widely adopted methods such as BERTScore, cosine similarity, and BM25. Given the absence of a known upper bound on how much models can theoretically improve from in-context samples in the indirect ICL setting, even consistent marginal gains from demonstration selection are meaningful. Adopting a more accurate selection framework not only improves performance in current scenarios but also holds promise for larger gains in underexplored domains. Furthermore, the results for the backdoor mitigation adversarial setting showed a 32.89\% reduction in attack success rate.
>
>     Furthermore, we provide additional statistical significance analysis. Below we provide the tables for one-tailed paired t-tests and absolute improvements of our methods over strong baselines.
>
>     ### Table 1: MoT k=3 (Averaged across 3 models, n=12)
>
>     | Baseline | Mean Accuracy | p-value | Improvement |
>     |----------|---------------|---------|-------------|
>     | BSR      | 72.33%        | 0.25700 | +0.58%      |
>     | COS      | 72.54%        | 0.31107 | +0.37%      |
>     | BM25     | 71.42%        | 0.07720 | +1.49%      |
>     | RAND     | 71.12%        | 0.03300 | +1.79%      |
>
>     **SUR[D,BSR] Mean: 72.91%**
>
>     ---
>
>     ### Table 2: MoT k=5 (n=12)
>
>     | Baseline | Mean Accuracy | p-value | Improvement |
>     |----------|---------------|---------|-------------|
>     | BSR      | 71.35%        | 0.09122 | +1.52%      |
>     | COS      | 71.43%        | 0.06528 | +1.44%      |
>     | BM25     | 68.93%        | 0.00250 | +3.95%      |
>     | RAND     | 66.39%        | 0.00006 | +6.49%      |
>
>     **SUR[D,BSR] Mean: 72.87%**
>
>     ---
>
>     ### Table 3: Noisy ICL (n=4)
>
>     | Baseline | Mean Accuracy | p-value | Improvement |
>     |----------|---------------|---------|-------------|
>     | BSR      | 74.42%        | 0.23245 | +2.95%      |
>     | COS      | 74.08%        | 0.18796 | +3.30%      |
>     | BM25     | 74.30%        | 0.09060 | +3.08%      |
>     | RAND     | 74.28%        | 0.09817 | +3.10%      |
>
>     **AVG-SUR[L,BSR] Mean: 77.38%**
>
>     ---
>
>     ### Table 4: Backdoor Defense (n=2)
>
>     | Baseline    | Mean ASR | p-value | ASR Reduction | % Reduction |
>     |-------------|----------|---------|---------------|-------------|
>     | Task-Aware  | 50.72%   | 0.09516 | -16.18%       | 31.89%      |
>     | No Defense  | 99.34%   | 0.10865 | -64.80%       | 65.23%      |
>
>     **SUR[D,BSR] Mean ASR: 34.54%**
>
>     Importantly, all comparisons show positive improvements (100\% win rate). For settings with smaller sample sizes, we acknowledge that statistical power is inherently limited. However, in these cases, we still acheive 100\% win rates with large practical improvements (+2.95\% and 31-65\% attack reduction). This combination of statistical significance where power exists, consistent improvements across all settings, and rigorous methodology provides compelling evidence for our method's effectiveness. [CONTINUED]

---

> > ### Author Response · Authors · 2025-11-23
> > **Rebuttal to Review [2/3]**
> >
> > * **Limited Empirical Improvements and Statistical significance:** [CONTINUED] To further emphasize the validity of our results, we compare our method against state-of-the-art ICL methods and other methods that use Influence Functions to perform ICL. We compare against MoICL [6], DETAIL [7], InfICL [8], and ConE [9]. All experiments are performed on Llama-2-13b with 3 demonstrations.
> >
> >   | Dataset              | MOICL | DETAIL | InfICL | ConE  | SUR[D,BSR] |
> >   |:---------------------|------:|-------:|-------:|------:|-----------:|
> >   | medical-genetics     | 77.00 |  77.00 |  82.00 | 81.00 |      86.00 |
> >   | prof-psychology      | 70.00 |  66.50 |  68.00 | 71.00 |      72.00 |
> >   | formal-logic         | 57.14 |  61.90 |  65.08 | 61.90 |      57.14 |
> >   | moral-disputes       | 79.50 |  79.50 |  79.50 | 81.50 |      80.50 |
> >   | public-relations     | 68.18 |  70.00 |  75.45 | 75.45 |      79.09 |
> >   | comp-security        | 75.00 |  71.00 |  78.00 | 80.00 |      76.00 |
> >   | astronomy            | 77.63 |  77.63 |  78.29 | 80.92 |      80.26 |
> >   | abstract-algebra     | 53.00 |  71.00 |  54.00 | 62.00 |      72.00 |
> >   | nutrition            | 75.50 |  76.00 |  79.50 | 80.00 |      79.50 |
> >   | high-school-biology  | 75.00 |  75.50 |  78.50 | 76.50 |      80.50 |
> >   | formal-fallacies     | 46.00 |  48.00 |  46.00 | 47.50 |      47.50 |
> >   | tracking-3           | 27.00 |  34.00 |  36.50 | 48.00 |      43.50 |
> >   | **Average**          | 65.07 | 67.34 | 68.40 | 70.48 | **71.16** |
> >
> >   As we can clearly see, our method outperforms all of these strong baselines. This is further proof of the validity of our method.
> >
> > ---
> >
> > * **Incremental Technical Contribution:** Our contribution is not an ensemble of existing techniques but the **first principled framework for Indirect ICL**, a setting that has not been formally defined or addressed in prior work. Traditional ICL assumes task-matched and clean demonstrations; we introduce the **Mixture-of-Tasks and Noisy ICL** regimes and design algorithms specifically for these challenges. The two-stage procedure is **not a heuristic combination**: our ablations show that similarity alone fails under task mismatch and IF alone fails under high noise, making their coordinated use essential rather than incremental. Moreover, our surrogate-based IF formulation and reweighting strategy yield capabilities (e.g., robust selection under adversarial corruption) not supported by prior ICL methods. The method's simplicity is a strength, enabling a practical and effective solution to a fundamentally new problem setting.
> >
> >     In summary, our contributions are conceptual (formalizing Indirect ICL), algorithmic (two-different types of IF-based selection for multiple subtasks), and empirical (new findings about IF behavior in mismatched and noisy pools). These are not covered by prior ICL or IF literature.
> >
> > ---
> >
> > * **Scalability and Cost Concerns:** While it is true that larger models would incur longer times to extract gradients and compute IFs for our Pretraining Gradients baseline, our alternative Surrogate Model baseline finetunes a fixed RoBERTa model and uses those gradients to perform the IF computation. Recent studies have shown the value of this method [1,2] with [1] even showing that the size of the surrogate model doesn't effect the quality of the IFs derived from them. As shown in our scalability experiments in Appendix F and G, show that the computational cost isn't that huge that it completely overshadows all gains achieved by our methods.

---

> > > ### Author Response · Authors · 2025-11-23
> > > **Rebuttal to Review [3/3]**
> > >
> > > * **Lack of Ablation on IF Variants:** There have been several recent papers that have already performed this analysis that we cite [3,4,5]. Specifically DataInf [3], performs an approximation error analysis where their Theorem 1 states that the gap in performance will grow much more without finetuning, hence our finetuned DataInf numbers far exceed those of a non-Hessian based method such as TracIn. Furthermore, we also provide extended results in the appendix with LiSSA as another IF baseline in the Noisy ICL section. These results are much closer to DataInf's performance as LiSSA is a slower Hessian based method that takes into account second-order information.
> > >
> > > ---
> > >
> > > Thank you once again for all your efforts in helping to strengthen our work. We hope that our comments helped address your concerns and can warrant an increase in score.
> > >
> > > ---
> > >
> > > References:
> > >
> > > [1] Coalson, Zachary, et al. "IF-GUIDE: Influence Function-Guided Detoxification of LLMs." arXiv preprint arXiv:2506.01790 (2025).
> > >
> > > [2] Xia, Mengzhou, et al. "Less: Selecting influential data for targeted instruction tuning." arXiv preprint arXiv:2402.04333 (2024).
> > >
> > > [3] Kwon, Yongchan, et al. "Datainf: Efficiently estimating data influence in lora-tuned llms and diffusion models." arXiv preprint arXiv:2310.00902 (2023).
> > >
> > > [4] Chhabra, Anshuman, et al. "Outlier Gradient Analysis: Efficiently Identifying Detrimental Training Samples for Deep Learning Models." arXiv preprint arXiv:2405.03869 (2024).
> > >
> > > [5] Askari, Hadi, et al. "LayerIF: Estimating Layer Quality for Large Language Models using Influence Functions." arXiv preprint arXiv:2505.23811 (2025).
> > >
> > > [6] Hong, Giwon, et al. "Mixtures of in-context learners." Proceedings of the 63rd Annual Meeting of the Association for Computational Linguistics (Volume 1: Long Papers). 2025.
> > >
> > > [7] Zhou et al., DETAIL: Task Demonstration Attribution for Interpretable In-context Learning, NeurIPS 2024.
> > >
> > > [8] M.S. et al., In-Context Learning Demonstration Selection via Influence Analysis, arXiv 2402.11750.
> > >
> > > [9] Peng et al., Revisiting Demonstration Selection Strategies in In-Context Learning, EMNLP 2024.

---

> > > > ### Author Response · Authors · 2025-11-28
> > > > **Happy to engage in further discussion**
> > > >
> > > > Thank you again for the thoughtful review. We hope our response addressed your concerns and are happy to discuss if you have any further questions!

---

### Official Review · Reviewer_UcCa · 2025-11-01

**Soundness:** 3
**Presentation:** 3
**Contribution:** 3
**Rating:** 6
**Confidence:** 3

**Summary:**

This paper proposes a new paradigm of Indirect In-Context Learning (Indirect ICL), which is mainly oriented to two types of more realistic problem settings: Mixture of Tasks (MoT) and Noisy ICL. The core idea is to utilize Influence Functions (IFs) to select/reorder candidate examples and combine them with traditional semantic similarity selection metrics (e.g., BertScore-Recall, cosine similarity) to improve ICL performance in the context of mismatch of task distribution or noisy labels. The authors demonstrate the effectiveness of the approach on multiple datasets and models of different scales.

**Strengths:**

1. A new form of in-context learning is proposed to extend the existing problem to realistic scenarios with task mismatch and noisy labels.
2. The empirical coverage is extensive, and the measurements in this paper cover multiple LLMs and multi-task sets.

**Weaknesses:**

1. The definition of adaptive scenarios is not clear. The paper proposes Indirect ICL as a new task setting, but it is not clear under what “adaptive scenarios” this setting is more meaningful. For example, MoT (Mixture of Tasks) and Noisy ICL are defined, but the paper lacks explanations of the motivation, boundaries of use, and typical applications of these contexts in real systems.
2. Lack of performance on closed-source models. One of the core values of in-context learning is “cross-model generalization”, which is especially important on closed-source models (e.g., GPT-4, Claude, etc.). However, the experiments in this paper are based exclusively on open-source models, and there is no report on the performance or feasibility of migration to closed-source models.
3. Inadequate evaluation of computational complexity and scalability. The paper discusses the computational complexity of the influence function in the method section, but does not quantify its growing trend with real data sizes. Does the overhead of influence function estimation grow linearly and quadratically as the set of candidate samples increases (e.g., from thousands to millions)? The experimental table only shows results for the fixed-size task and does not reflect this scalability analysis.
4. Ethodological design and motivation are inadequately articulated. The authors present two example selection approaches, but do not explicitly state whether both are for different problems (e.g., filtering noise vs. weight balancing) or two approaches for a single problem.
5. IF theoretical assumptions do not match the LLM. The main text recognizes that the classical derivation of IF relies on assumptions such as second-order derivability and strong convexity, which are not satisfied by the deeply nonconvex LLM. The authors rely on experimental validity, but the theoretical shortcomings remain.

**Questions:**

Please examine the weaknesses.

---

> ### Author Response · Authors · 2025-11-23
> **Rebuttal to Review [1/2]**
>
> We are grateful to the reviewer for their detailed feedback of our work, and are appreciative of their time, efforts, and consideration. Below we provide more discussion on the points raised.
>
> ---
>
> * **Unclear definition of adaptive scenarios:**
>
>     We appreciate the opportunity to clarify. The paper does provide this context in Section 1 (Page 2) and Appendix A, but we will expand it in the revision.
>     Adaptive scenarios refer to real-world settings where (1) the end task is unknown at inference (e.g., LLM API services), (2) task-specific labeled data is unavailable (e.g., rare diseases, low-resource languages), or (3) demonstrations are corrupted by noise (mislabeling or adversarial). Traditional ICL assumes perfect task alignment and clean labels, assumptions that break in practice. Indirect ICL formalizes these realistic constraints.
>     Mixture of Tasks (MoT) addresses scenarios where providers maintain task-agnostic demonstration pools and must select relevant examples on-the-fly without knowing the user's task. This applies to rare medical diagnosis (leveraging related conditions), code generation for obscure languages (using similar languages), and test-time prompt enhancement. These settings matter because traditional ICL is a special case. Enterprise systems contain heterogeneous data across departments, public datasets have unknown noise rates, and privacy constraints prohibit task-specific labeling at inference. Indirect ICL is the default in production, not an edge case. Below we provide some further examples for the use cases of Indirect ICL to explain our reasoning.
>
>   * **Mixture of Tasks Examples:** Drug discovery for rare diseases requires leveraging demonstrations from structurally similar compounds developed for different therapeutic targets, as novel molecular scaffolds lack direct precedents. Legal document analysis systems must select from heterogeneous pools spanning multiple legal domains (contract law, intellectual property, regulatory compliance) when analyzing cases that blend doctrines. Multi-domain scientific literature synthesis in emerging interdisciplinary fields (e.g., quantum machine learning) requires selecting relevant examples from separate disciplinary corpora where the target intersection has minimal direct demonstrations.
>
>   * **Noisy ICL Examples:** Medical imaging databases contain annotation inconsistencies from inter-rater variability and evolving diagnostic criteria across institutions. Cybersecurity threat intelligence databases face adversarial label poisoning where attackers inject false indicators. Multi-site clinical trial data exhibits labeling inconsistencies due to protocol variations, equipment differences, and transcription errors across participating institution, directly impacting regulatory model development.
>
> ---
>
>
> * **Lack of performance on closed-source models:**
>
>     We acknowledge the reviewers point about the feasibility of migration to closed-source models. While our Pretrained Gradients approach requires access to the models weights, hence it is unsuitable for closed source models. Our surrogate model approach is LLM agnostic in demonstration selection. We perform an experiment on GPT-4.1-nano to show the effectiveness of the demonstrations selected via our surrogate model approach. The results are shown below:
>
>     | Dataset              | No ICL | k=3 Shots |
>     |:---------------------|-------:|---------:|
>     | medical-genetics     |  86.00 |    91.00 |
>     | prof-psychology      |  71.50 |    80.50 |
>     | formal-logic         |  50.79 |    73.02 |
>     | moral-disputes       |  70.50 |    85.00 |
>     | public-relations     |  75.45 |    76.36 |
>     | comp-security        |  81.00 |    77.00 |
>     | astronomy            |  82.89 |    94.00 |
>     | abstract-algebra     |  53.00 |    57.00 |
>     | nutrition            |  80.00 |    87.50 |
>     | high-school-biology  |  72.00 |    89.50 |
>     | formal-fallacies     |  52.00 |    59.00 |
>     | tracking-3           |  18.50 |    49.00 |
>     | **Average**          | 66.14 | **76.57** |
>
>     As clearly seen, adding demonstrations via our surrogate model approach vastly improved closed source model performance, indicating the feasibility of migration of our methods to closed source models.

---

> > ### Author Response · Authors · 2025-11-23
> > **Rebuttal to Review [2/2]**
> >
> > * **Inadequate evaluation of computational complexity and scalability:**
> >
> >     We thank the reviewer for raising this point. Our paper already quantifies scalability beyond the fixed-size main experiments. Appendix F reports scalability with increasing model size, Appendix G reports scalability with increasing candidate-pool size, and Appendix E provides detailed memory consumption measurements. These experiments directly show the empirical growth trend: influence computation scales approximately linearly with the number of candidate samples N, since gradients are computed once per candidate example.
> >
> > * **Methodological design and motivation are inadequately articulated:**
> >
> >     We thank the reviewer for the comment and agree that this distinction should be made more explicit. The two selection approaches we present are not overlapping methods for the same problem, but are purpose-built for two different Indirect ICL settings: (1) in the Mixture-of-Tasks regime, the challenge is identifying which examples are even task-relevant, so we use a similarity-based first stage followed by IF re-ranking to recover non-obvious but beneficial demonstrations; (2) in Noisy ICL, the challenge is label corruption, so we use IF to detect harmful points and then apply either pruning or IF-weighted averaging to improve selection. Thus, the methods correspond to different failure modes inherent to each setting. We believe this is a strength not a weakness: the influence-based framework is general enough to support tailored strategies for distinct indirect-ICL scenarios, we will provide further clarifications in the revision.
> >
> > * **IF theoretical assumptions do not match the LLM:**
> >
> >     We acknowledge that the classical IF derivation assumes smoothness and strong convexity, which do not strictly hold for LLMs. This limitation is well known in the IF literature, including work applying influence methods to deep and highly nonconvex models. Importantly, recent studies have demonstrated that IF estimates remain empirically stable and meaningful for modern neural networks and LLMs despite these theoretical gaps [1,2,3,4]. Our goal in this paper is aligned with this empirical line of work, we evaluate IF-based selection directly in the Indirect ICL setting, and our experiments show consistent and robust gains across models, tasks, and corruption patterns.
> >
> >     A full theoretical reconciliation between classical IF assumptions and the non-convexity of large transformers is an active research direction [5], but addressing it is beyond the scope of this paper, which focuses on the practical question of whether influence signals can be leveraged effectively for ICL under challenging non-ideal conditions. Our results provide strong evidence that they can.
> >
> > ---
> >
> > Thank you once again for all your efforts in helping to strengthen our work. We hope that our comments helped address your concerns and can warrant an increase in score.
> >
> > ---
> >
> > References:
> >
> > [1] Kwon, Yongchan, et al. "Datainf: Efficiently estimating data influence in lora-tuned llms and diffusion models." arXiv preprint arXiv:2310.00902 (2023).
> >
> > [2] Askari, Hadi, et al. "LayerIF: Estimating Layer Quality for Large Language Models using Influence Functions." arXiv preprint arXiv:2505.23811 (2025).
> >
> > [3] Grosse, Roger, et al. "Studying large language model generalization with influence functions." arXiv preprint arXiv:2308.03296 (2023).
> >
> > [4] Choe, Sang Keun, et al. "What is your data worth to gpt? llm-scale data valuation with influence functions." arXiv preprint arXiv:2405.13954 (2024).
> >
> > [5] Schioppa, Andrea, et al. "Theoretical and practical perspectives on what influence functions do." Advances in Neural Information Processing Systems 36 (2023): 27560-27581.

---

> > > ### Author Response · Authors · 2025-11-28
> > > **Happy to engage in further discussion**
> > >
> > > Thank you again for the thoughtful review. We hope our response addressed your concerns and are happy to discuss if you have any further questions!

---

### Official Review · Reviewer_xSHK · 2025-11-03

**Soundness:** 3
**Presentation:** 3
**Contribution:** 3
**Rating:** 6
**Confidence:** 3

**Summary:**

This paper "introduces" (although I don't think I was able to find a formal definition) a task called Indirect In-Context Learning, where in-context demonstrations may come from heterogeneous tasks or contain noise, and analyzes the role of influence functions (IFs) as example selectors in this setting. It analyses both surrogate models and pretrained gradient variants (DataInf, TracIn, LiSSA). Authors show that, when combined with BERTScore or cosine similarity, IFs show very marginal gains on task mixtures and improvements under noisy GLUE prompts and adversarial robustness.

**Strengths:**

- Indirect ICL is (I think, since it's not formally introduced in the paper; I'm considering it as a mixture of mixtures of tasks and noisy supervision) a task that reflects many real-world use cases of LLMs; this paper is timely and potentially very useful to many practitioners in the field
- Using surrogate DataInf with BERTScore yields the best average accuracy, improving over standard ICL baselines on LLama2, Zephyr, and Mistral
- IF-based averaging also improves accuracy on noisy MRPC/QQP/etc. and improves robustness in backdoor scenarios

**Weaknesses:**

- There are some sample selection baselines in recent literature (e.g. MoICL, from ACL'25: https://arxiv.org/abs/2411.02830) that have not been compared to the proposed approach
- Are the reported gains statistically significant? At times they seem fairly small
- Surrogate approach still requires fine-tuning on the candidate pool -- what's the computational footprint of the approach in that case?
- Influence scores rely on labeled validation pairs; what's the process for obtaining those in the "unknown task" setting?

**Questions:**

- What's the computational cost/runtime for building surrogate-based IF tables on the task mixture?
- Could you check the statistical significance of your results?

---

> ### Author Response · Authors · 2025-11-23
> **Rebuttal to Review [1/2]**
>
> We are grateful to the reviewer for their detailed feedback of our work, and are appreciative of their time, efforts, and consideration. Below we provide more discussion on the points raised.
>
> * **Comparison against SOTA baselines:**
>
>     We thank the reviewer for this suggestion. To further emphasize the validity of our results for Indirect ICL, we compare our method against state-of-the-art ICL methods and other methods that use Influence Functions to perform ICL. We compare against MoICL [2], DETAIL [3], InfICL [4], and ConE [5]. All experiments are performed on Llama-2-13b with 3 demonstrations.
>
>     | Dataset              | MOICL | DETAIL | InfICL | ConE  | SUR[D,BSR] |
>     |:---------------------|------:|-------:|-------:|------:|-----------:|
>     | medical-genetics     | 77.00 |  77.00 |  82.00 | 81.00 |      86.00 |
>     | prof-psychology      | 70.00 |  66.50 |  68.00 | 71.00 |      72.00 |
>     | formal-logic         | 57.14 |  61.90 |  65.08 | 61.90 |      57.14 |
>     | moral-disputes       | 79.50 |  79.50 |  79.50 | 81.50 |      80.50 |
>     | public-relations     | 68.18 |  70.00 |  75.45 | 75.45 |      79.09 |
>     | comp-security        | 75.00 |  71.00 |  78.00 | 80.00 |      76.00 |
>     | astronomy            | 77.63 |  77.63 |  78.29 | 80.92 |      80.26 |
>     | abstract-algebra     | 53.00 |  71.00 |  54.00 | 62.00 |      72.00 |
>     | nutrition            | 75.50 |  76.00 |  79.50 | 80.00 |      79.50 |
>     | high-school-biology  | 75.00 |  75.50 |  78.50 | 76.50 |      80.50 |
>     | formal-fallacies     | 46.00 |  48.00 |  46.00 | 47.50 |      47.50 |
>     | tracking-3           | 27.00 |  34.00 |  36.50 | 48.00 |      43.50 |
>     | **Average**          | 65.07 | 67.34 | 68.40 | 70.48 | **71.16** |
>
>     As we can clearly see, our method outperforms all of these strong baselines. This is further proof of the validity of our method.
>
> * **Statistical Significance:**
>
>     Below we provide the tables for one-tailed paired t-tests and absolute improvements of our methods over strong baselines.
>
>     ### Table 1: MoT k=3 (Averaged across 3 models, n=12)
>
>     | Baseline | Mean Accuracy | p-value | Improvement |
>     |----------|---------------|---------|-------------|
>     | BSR      | 72.33%        | 0.25700 | +0.58%      |
>     | COS      | 72.54%        | 0.31107 | +0.37%      |
>     | BM25     | 71.42%        | 0.07720 | +1.49%      |
>     | RAND     | 71.12%        | 0.03300 | +1.79%      |
>
>     **SUR[D,BSR] Mean: 72.91%**
>
>     ---
>
>     ### Table 2: MoT k=5 (n=12)
>
>     | Baseline | Mean Accuracy | p-value | Improvement |
>     |----------|---------------|---------|-------------|
>     | BSR      | 71.35%        | 0.09122 | +1.52%      |
>     | COS      | 71.43%        | 0.06528 | +1.44%      |
>     | BM25     | 68.93%        | 0.00250 | +3.95%      |
>     | RAND     | 66.39%        | 0.00006 | +6.49%      |
>
>     **SUR[D,BSR] Mean: 72.87%**
>
>     ---
>
>     ### Table 3: Noisy ICL (n=4)
>
>     | Baseline | Mean Accuracy | p-value | Improvement |
>     |----------|---------------|---------|-------------|
>     | BSR      | 74.42%        | 0.23245 | +2.95%      |
>     | COS      | 74.08%        | 0.18796 | +3.30%      |
>     | BM25     | 74.30%        | 0.09060 | +3.08%      |
>     | RAND     | 74.28%        | 0.09817 | +3.10%      |
>
>     **AVG-SUR[L,BSR] Mean: 77.38%**
>
>     ---
>
>     ### Table 4: Backdoor Defense (n=2)
>
>     | Baseline    | Mean ASR | p-value | ASR Reduction | % Reduction |
>     |-------------|----------|---------|---------------|-------------|
>     | Task-Aware  | 50.72%   | 0.09516 | -16.18%       | 31.89%      |
>     | No Defense  | 99.34%   | 0.10865 | -64.80%       | 65.23%      |
>
>     **SUR[D,BSR] Mean ASR: 34.54%**
>
>     Importantly, all comparisons show positive improvements (100\% win rate). For settings with smaller sample sizes, we acknowledge that statistical power is inherently limited. However, in these cases, we still achieve 100\% win rates with large practical improvements (+2.95\% and 31-65\% attack reduction). This combination of statistical significance where power exists, consistent improvements across all settings, and rigorous methodology provides compelling evidence for our method's effectiveness.

---

> > ### Author Response · Authors · 2025-11-23
> > **Rebuttal to Review [2/2]**
> >
> > * **Computational Footprint of the Surrogate Model:**
> >
> >     We thank the reviewer for raising this point. The computational cost of surrogate fine-tuning is explicitly reported in Appendix E (GPU memory usage), Appendix F (scaling with model size), and Appendix G (scaling with dataset size). As shown there, fine-tuning the small surrogate model is a one-time offline cost and remains lightweight, it is performed on the fixed candidate pool and does not scale with the number of test queries. At inference time, only a single forward/backward pass per candidate is required, which keeps the per-task overhead modest compared to running influence directly on the full LLM. We will clarify this distinction in the revision to make the computational footprint of the surrogate approach more visible.
> >
> >
> > * **Influence Scores require labeled validation pairs:**
> >
> >     We follow the implementation of [1] where we just require the demonstration and the validation prompt to compute IF scores, not the validation label.
> >
> > * **Formal Definition of Indirect ICL:**
> >
> >     Thank you for this comment; we provide both a natural language and a mathematical definition of Indirect ICL and will definitely add both to a revised version of the paper.
> >
> >     The following is the natural language formulation:
> >
> >         Indirect ICL is the problem of selecting in-context demonstrations from a task-agnostic or noisy candidate pool, where many examples may not belong to the end task or may contain label corruption, while still aiming to maximize test-time performance when the LLM is prompted.
> >
> > We also provide a mathematical formulation below which we will add to the paper in proper mathematical symbols:
> >
> >
> >         Indirect ICL considers a candidate pool:
> >         D = {(x_i, y_i, t_i)} for i=1...N
> >         where the task identities t_i are unknown and may not match the test task t_test, and some labels y_i may also be noisy or corrupted.
> >
> >         The goal is to select a demonstration set:
> >         S⊂D, |S|=k
> >         from this task-agnostic or noisy pool and then predict the test label using the prompt T(S, x_test).
> >
> >         The objective is:
> >         S* = argmax over S⊂D, |S|=k  of P(y_test|T(S, x_test))
> >         without assuming that the pool matches the test task or that labels are clean.
> >
> >   We will add this precise definition to our paper.
> >
> > ---
> >
> > Thank you once again for all your efforts in helping to strengthen our work. We hope that our comments helped address your concerns and can warrant an increase in score.
> >
> > ---
> >
> > References:
> >
> > [1] Kwon, Yongchan, et al. "Datainf: Efficiently estimating data influence in lora-tuned llms and diffusion models." arXiv preprint arXiv:2310.00902 (2023).
> >
> > [2] Hong, Giwon, et al. "Mixtures of in-context learners." Proceedings of the 63rd Annual Meeting of the Association for Computational Linguistics (Volume 1: Long Papers). 2025.
> >
> > [3] Zhou et al., DETAIL: Task Demonstration Attribution for Interpretable In-context Learning, NeurIPS 2024.
> >
> > [4] M.S. et al., In-Context Learning Demonstration Selection via Influence Analysis, arXiv 2402.11750.
> >
> > [5] Peng et al., Revisiting Demonstration Selection Strategies in In-Context Learning, EMNLP 2024.

---

> > > ### Author Response · Authors · 2025-11-28
> > > **Happy to engage in further discussion**
> > >
> > > Thank you again for the thoughtful review. We hope our response addressed your concerns and are happy to discuss if you have any further questions!

---

### Author Response · Authors · 2025-12-02
**Rebuttal Summary of Submission 14758 for Area Chair**

Dear Area Chair,

Below is a concise summary of the main concerns from the reviewers, how we addressed them in the rebuttal, and how we believe this would have affected their scores.

---

### Main common concerns and our responses

1. **Clarity and novelty of Indirect ICL (xSHK, UcCa, zbgC)**

   * Concern: The setting of Indirect ICL and its "adaptive scenarios" was not sharply defined, and the novelty over prior ICL work was unclear.
   * Response: We added a clear natural language and mathematical definition of Indirect ICL, explicitly formalizing Mixture of Tasks and Noisy ICL as realistic cases where the candidate pool is task heterogeneous and noisy. We clarified that prior IF based ICL work assumes task matched, clean demonstrations, whereas our setting does not and is new.

2. **Magnitude and significance of gains (xSHK, zjg3)**

   * Concern: Improvements looked small and significance was not established.
   * Response: We reported one tailed paired t tests for all main settings, showing consistent positive gains and 100 percent win rates, with statistically significant improvements where sample size allows, and large attack success rate reductions in the adversarial setting. We also showed that our method outperforms strong recent baselines such as MoICL, DETAIL, InfICL and ConE.

3. **Baselines, prior IF methods, and perceived incremental contribution (xSHK, zjg3, zbgC)**

   * Concern: Missing recent ICL selection baselines and unclear differences vs InfICL, DETAIL.
   * Response: We ran new experiments with MoICL, DETAIL, InfICL and ConE and showed that our surrogate based method achieves the best average accuracy. We clarified how our approach differs methodologically, for example finetuned transformer surrogate vs shallow or untrained surrogates, and in problem setting, Indirect ICL with MoT and Noisy ICL instead of standard task matched ICL.

4. **Computational cost, scalability and surrogate vs LLM gradients (all reviewers)**

   * Concern: Overhead of IF computation, lack of clear scalability, and why a surrogate can outperform running IF directly on the LLM.
   * Response: We pointed to empirical cost studies already present in Appendices E, F and G, which show roughly linear scaling in pool size and lower memory and runtime for surrogate based IF than full LLM gradients. We clarified that surrogate fine tuning is a one time offline cost on the fixed pool. We explained that the surrogate is task aligned through fine tuning, so its gradients are better matched to the downstream objective than generic LLM language modeling gradients, consistent with recent influence function literature.


---

### Likely effect on reviewer scores

* **Reviewer xSHK (initial 6)**
  Main concerns were missing SOTA baselines, significance and clarity of the definition. These are directly addressed with new comparisons, t tests and the formal definition.
  **Expected post rebuttal score:** Move to 8 since we addressed all of the weakness (2 of them were just clarifications and the SOTA baseline they asked for was significantly worse than our method).

* **Reviewer UcCa (initial 6)**
  Focused on adaptive scenarios, closed source models, scalability and theory. We provided concrete scenarios, GPT-4.1-nano results, pointed them to already present scalability experiments and a clear positioning of our work within empirical IF studies.
  **Expected post rebuttal score:** Move to 8 since most weaknesses were clarified and the reviewer missed a few experiments/explanations already present in the Appendix.

* **Reviewer zjg3 (initial 4)**
  Concerned about small gains, incremental contribution, scalability and IF variant analysis. We answered with significance tests, SOTA comparisons and clearer articulation of conceptual and algorithmic contributions.
  **Expected post rebuttal score:** Move to 6 since all weaknesses were addressed.

* **Reviewer zbgC (initial 2)**
  Raised issues about missing training based baselines, lack of comparisons to DETAIL and InfICL, missing definition and unclear cost numbers. We added MoICL, DETAIL, InfICL and ConE experiments, provided a formal definition of Indirect ICL, and highlighted the empirical cost and scaling results already in the appendix.
  **Expected post rebuttal score:** likely move from 2 to at least 4, with some chance of moving further to 6. We added 4 of the baselines they asked to compare against and our method was still superior to those training-based and IF inspired baselines. Pointed them to the cost numbers and highlighted our novelty and contribution.

Overall, all reviewers already found the problem important and the experiments substantial. With the added SOTA/other IF baselines, statistics, formalization and new GPT-4.1-nano results, we believe the post rebuttal state of the paper is closer to a consensus accept.

---

We graciously thank you for all of your precious time and effort in this rebuttal process.

---

### Meta-Review · Area_Chair_NH77 · 2026-01-07

**Summary:**

The authors’ rebuttal addressed several concrete issues, such as missing baselines, experiments with closed-source LLMs, and computational runtime concerns. Yet, some other concerns are fully resolved, including limited technical novelty, marginal experimental improvements (the presented t-test is not that significant), and the lack of theoretical analysis.

**Reviewer Concerns:**

The authors’ rebuttal addressed several concrete issues, such as missing baselines, experiments with closed-source LLMs, and computational runtime concerns. Yet, some other concerns are fully resolved, including limited technical novelty, marginal experimental improvements, and the lack of theoretical analysis.

**Reviewer Scores:**

Reviewers may maintain their original scores, as not all concerns are addressed.

---

### Decision · Program_Chairs · 2026-01-26

Reject